# Discovery of type II polyketide synthase-like enzymes for the biosynthesis of cispentacin

Genki Hibi [1], Taro Shiraishi [1,2], Tatsuki Umemura[1], Kenji Nemoto [1], Yusuke Ogura [1], Makoto Nishiyama [1,2] & Tomohisa Kuzuyama [1,2] ✉

Type II polyketide synthases (PKSs) normally synthesize polycyclic aromatic compounds in nature, and the potential to elaborate further diverse skeletons was recently revealed by the discovery of a polyene subgroup. Here, we show a type II PKS machinery for the biosynthesis of a five-membered nonaromatic skeleton contained in the nonproteinogenic amino acid cispentacin and the plant toxin coronatine. We successfully produce cispentacin in a heterologous host and reconstruct its biosynthesis using seven recombinant proteins in vitro. Biochemical analyses of each protein reveal the unique enzymatic reactions, indicating that a heterodimer of type II PKS-like enzymes (AmcF–AmcG) catalyzes a single $C_2$ elongation as well as a subsequent cyclization on the acyl carrier protein (AmcB) to form a key intermediate with a five-membered ring. The subsequent reactions, which are catalyzed by a collection of type II PKS-like enzymes, are also peculiar. This work further expands the definition of type II PKS and illuminates an unexplored genetic resource for natural products.

Polyketide synthases (PKSs) are multifunctional enzymes that produce diverse natural products, which often exhibit impressive biological activities. Of these PKS enzymes, subclass type II PKSs synthesize a variety of aromatic polycyclic polyketides, such as anthracyclines and tetracyclines. The typical type II PKSs contain ketosynthase (KS), which forms a heterodimer with its chain length factor (CLF) partner. This KS–CLF heterodimer regulates the number of condensation reactions occurring during the elongation of the polyketide chain, which is exclusively catalyzed on the acyl carrier protein (ACP). The poly-β-keto intermediates produced are subsequently converted into the frameworks of aromatic polyketides through cyclization catalyzed by modifying enzymes, such as a ketoreductase (KR) and an aromatase/cyclase (Supplementary Fig. 1). A highly reducing (HR) type II PKS subfamily was recently discovered to produce polyenes. The type II PKS subfamily comprises a KS–CLF heterodimer, a KR, and a dehydratase (DH), which catalyze the repetitive cycles of chain elongation, β-ketoreduction, and dehydration, respectively, to produce aliphatic polyene structures (Supplementary Fig. 1). Thus, the diversity of known PKS and the presence of numerous uncharacterized PKS in genome databases indicate that PKS is an excellent source for skeletal formations.

Cispentacin ((1R,2S)−2-aminocyclopentane-1-carboxylic acid) consists of a carboxy group and an amino group that are attached to the beta position of a five-membered ring. This nonproteinogenic amino acid was isolated from *Bacillus cereus* L450-B2 and *Streptomyces setonii* No. 7562 and strongly inhibits the growth of *Candida albicans*[1,2]. BAY 10-8888, an analog of cispentacin, competitively inhibits isoleucyl-tRNA synthetase, thereby inhibiting protein synthesis and exhibiting anti-Candida activity[3–5]. Thus, cispentacin and its analogs have been regarded as promising pharmacological agents, and many efforts have been made to prepare them from both chemical and biosynthetic aspects.

Cispentacin is a partial structure of the peptidyl nucleoside antibiotic amipurimycin produced by *Streptomyces* bacterial strains (Fig. 1). We have identified the biosynthetic gene cluster (BGC) for amipurimycin (*amc*) in *Streptomyces* sp. SN-C1[6]. Bioinformatics analysis of the *amc* cluster led us to a gene cluster consisting of *cfa1, cfa2, cfa3, cfa4,* and *cfa5* in the plant pathogenic bacterium *Pseudomonas*

[1]Graduate School of Agricultural and Life Sciences, The University of Tokyo, 1-1-1 Yayoi, Bunkyo-ku, Tokyo 113-8657, Japan. [2]Collaborative Research Institute for Innovative Microbiology, The University of Tokyo, 1-1-1 Yayoi, Bunkyo-ku, Tokyo 113-8657, Japan. ✉e-mail: utkuz@g.ecc.u-tokyo.ac.jp

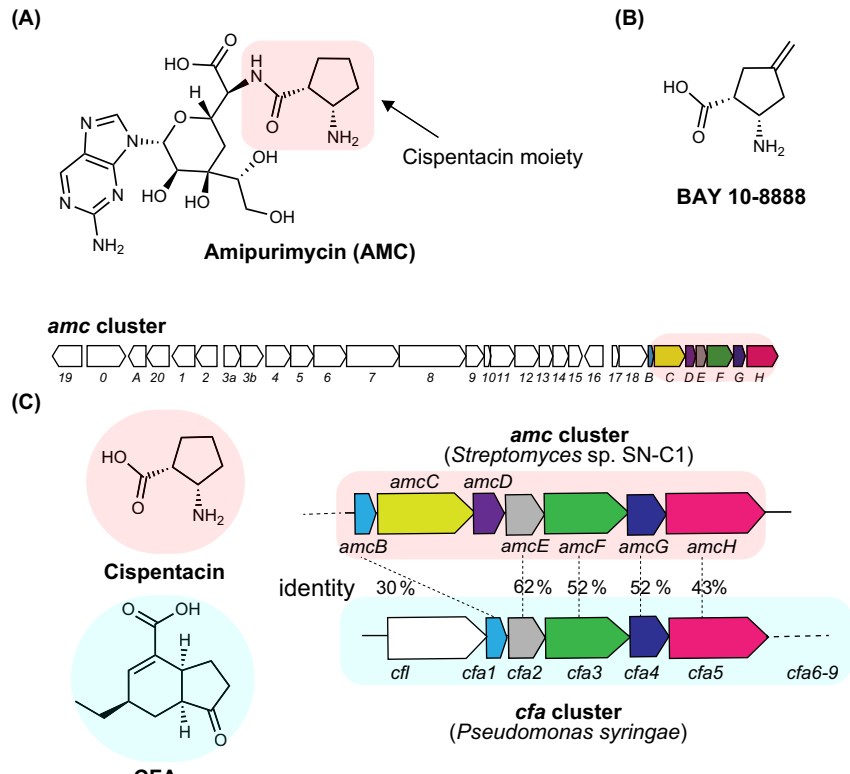

**Fig. 1 | Overview of cispentacin and its biosynthetic genes. A** Structure of amipurimycin and the biosynthetic gene cluster for amipurimycin (*amc* cluster). The highlighted genes are presumed to be involved in cispentacin biosynthesis. **B** The structure of BAY 10-8888 ((1 *R*,2 *S*)−2-amino-4-methylenecyclopentane-1-carboxylic acid), an antifungal analog of cispentacin. **C** Comparison of the gene clusters for cispentacin and CFA.

*syringae* (Fig. 1). The *cfa* genes are putative biosynthetic genes for coronafacic acid (CFA), which contains a five-membered ring similar to cispentacin and a partial structure of the phytotoxin coronatine[7,8]. Previous bioinformatics studies have postulated that the CFA biosynthetic pathway involves the type II PKS-like enzymes Cfa1 (ACP), Cfa2 (DH), Cfa3 (KS), Cfa4 (unknown function), and Cfa5 (acyltransferase; AT). The hypothetical pathway begins with the conversion of succinic semialdehyde derived from α-ketoglutarate (2-OG) into its CoA ester by Cfa5. Cfa1 (ACP), Cfa3 (KS), and Cfa4 have been postulated to produce Cfa1-bound 2-hydroxy-5-oxocyclopentane-1-carboxylic acid, which is subsequently dehydrated by Cfa2 (DH) to give Cfa1-bound 5-oxocyclopent-1-ene-1-carboxylic acid (Supplementary Fig. 2). The deduced amino acid sequences of *amcB*, *amcE*, *amcF*, *amcG*, and *amcH* show high similarity to those of *cfa1*, *cfa2*, *cfa3*, *cfa4*, and *cfa5*, respectively (Fig. 1), which suggests that the biosynthesis of CFA and cispentacin shares a common route leading to the pivotal intermediate 5-oxocyclopent-1-ene-1-carboxylic acid (**3**) bound to ACP (Supplementary Fig. 2)[6,9]. In addition, feeding studies using [13]C-labeled precursors have suggested that the starter substrate is 2-OG and that the C-1 carbon from 2-OG must be eliminated during the biosynthesis of **3**-ACP[6,8,10]. However, the biosynthetic route leading to the key intermediate **3**-ACP has not been documented, and the functions of each *amc* gene or *cfa* gene remain to be biochemically elucidated.

In this work, we successfully produced cispentacin in the heterologous host *Streptomyces albus* and reconstituted the biosynthesis of cispentacin in vitro, allowing us to establish the complete biosynthetic pathway of cispentacin. The establishment of cispentacin biosynthesis has resulted in the identification of a phylogenetically independent subfamily of type II PKSs and significantly rewrote the previously postulated biosynthetic mechanism of pivotal intermediate **3** of cispentacin and CFA. This work could further revise our definition

of type II PKS enzymes, which was recently revised with the elucidation of HR type II PKS pathways.

## Results

### Heterologous production of cispentacin and identification of the BGC of cispentacin

As *amcB*, *amcE*, *amcF*, *amcG*, and *amcH* likely form a single operon together with *amcC* and *amcD* (Fig. 1 and Supplementary Table 1 for putative functions of each gene), we hypothesized that these seven genes presumably function in a cooperative manner. We thus performed heterologous expression experiments, expecting that the introduction of the *amcB–amcH* genes into a cispentacin nonproducing *S. albus* G153 leads to the production of cispentacin or its biosynthetic intermediate(s). The resulting *S. albus* G153/pSEamcB–H transformant harboring *amcB–H* produced an average of 200 mg/L cispentacin (Supplementary Fig. 3, Supplementary Fig. 4, and Supplementary Fig. 5), demonstrating that the seven gene regions spanning *amcB* to *amcH* are responsible for cispentacin biosynthesis.

### Elucidation of the biosynthetic pathway of 3-AmcB

To reconstitute the biosynthesis of cispentacin in vitro, we prepared seven recombinant proteins, AmcB to AmcH. Soluble recombinant AmcF, AmcG, and AmcH proteins were purified from *S. albus* G153, and the others were purified from *Escherichia coli*. Interestingly, AmcG was insoluble when expressed alone in *S. albus*, whereas when expressed together with AmcF in *S. albus*, it became soluble and formed a heterodimer with AmcF (Supplementary Figs. 6, 7).

AmcB is assigned as ACP, on which chain elongation reactions in both fatty acid and polyketide biosynthesis are always catalyzed. For AmcB to function as an ACP, an inactive form of ACP (*apo*-AmcB) must undergo posttranslational modification, where *apo*-AmcB is converted to an active form (*holo*-AmcB) with a 4-phosphopantetheinyl group

attached to the serine residue of AmcB. Therefore, *holo*-AmcB was prepared by coexpression in *E. coli* with the *pptA2* gene encoding 4-phosphopantetheinyl transferase[11]. LC-electrospray ionization (ESI)-high-resolution mass spectrometry (HRMS) analysis of recombinant AmcB clearly showed the production of *holo*-AmcB (11681 Da) (Supplementary Fig. 8). It was also confirmed that the resulting *holo*-AmcB was converted to malonyl-AmcB (11767 Da) in the presence of malonyl-CoA in vitro (Supplementary Fig. 8). Malonyl-ACP is a typical $C_2$ elongation substrate for KS in the type II PKS system. Malonyl transfer to ACP has been validated with the AmcB homolog Cfa1, and FAS malonyl CoA:ACP transacylase (FabD), which is recruited to Cfa1, catalyzes the reaction[12]. Thus, with respect to AmcB, the malonyl transfer to AmcB may also be catalyzed by FabD, which is presumably recruited to AmcB.

As AmcH is assigned as an adenylate-forming enzyme, to identify a substrate to be adenylated by AmcH, we incubated AmcH with possible substrates (2-oxoglutarate, L-glutamate, L-aspartate, citrate, malate, fumarate, and succinate) in the presence of ATP at 30 °C for 1 h. LC-ESI-HRMS analysis of the reaction mixture clearly showed that AmcH loads only 2-OG onto AmcB to form 2-oxoglutaryl-AmcB (**1**-AmcB; 11809 Da) (Supplementary Fig. 9 and upper chromatogram (− AmcFG) in Fig. 2A). In addition, HPLC analysis showed that the consumption of ATP and the generation of AMP occurred in both AmcB- and 2-OG-dependent manners (Supplementary Fig. 10). These results demonstrate that AmcH functions as a 2-oxoglutaryl-ACP ligase, which adenylates 2-OG to temporarily form 2-oxoglutaryl-AMP and then transfers the 2-oxoglutaryl moiety from 2-oxoglutaryl-AMP to ACP to form 2-oxoglutaryl-ACP. AmcH is adenylate-forming acyltransferase (abbreviated as "A") that forms 2-oxoglutaryl-ACP. The identification of 2-OG as the substrate of AmcH suggests that **1**-AmcB is a starter substrate of the AmcF−AmcG heterodimer, which is consistent with previous feeding studies of the biosynthesis of cispentacin[6] and coronafacic acid[9,10].

Having obtained *holo*-AmcB and identified the AmcH substrate, we incubated *holo*-AmcB and AmcH with the AmcF−AmcG heterodimer and substrates (2-OG as the substrate of AmcH, ATP as a cofactor of AmcH, and malonyl-CoA as an elongation substrate of KS) at 30 °C for 1 h (designated Reaction 1 mixture). The reaction products were analyzed by LC-ESI-HRMS (see SI for LC conditions). A clear signal (11833.1 Da) was detected, corresponding to the mass of Compound **2** tethered to AmcB (**2**-AmcB) (Fig. 2A). To further confirm the molecular structure of **2**, AmcB was digested by thermolysin and then analyzed with LC-ESI-HRMS. The analysis confirmed $m/z$ 826.2567 $[M + H]^+$, corresponding to the mass of the *holo*-AmcB peptide fragment $^{36}$LDS$^{38}$ with a phosphopantetheine arm acylated with **2** with an error of 0.001 Da (Supplementary Fig. 11). Furthermore, a tandem MS-based phosphopantetheinyl ejection assay (PPant ejection assay)[13] determined the exact mass and MSMS fragment of **2** to indicate **2** as 3-oxocyclopent-1-ene-1,2-dicarboxylic acid (Fig. 2B and Supplementary Fig. 12). Finally, the structure of **2** was substantiated, as the retention time and exact mass in HR-LC–MS/MS analysis well corresponded to those of the chemically synthesized 3-oxocyclopent-1-ene-1,2-dicarboxylic acid standard (Figs. 2C, D).

These results demonstrate that the AmcF−AmcG heterodimer accepts **1**-AmcB as a starter substrate to catalyze a single $C_2$ elongation using malonyl-AmcB and subsequent cyclization concomitant with dehydration to form **2**-AmcB. This series of reactions involves unprecedented biosynthetic mechanisms catalyzed by type II PKS-like enzymes to form a five-membered ring. The heterodimer is also an unusual set of complexes. The typical type II PKS forms a heterodimer consisting of a KS and a specific CLF. This KS−CLF heterodimer regulates the number of condensation reactions occurring during the elongation of the polyketide chain, which is exclusively catalyzed on ACP. AmcF exhibits significant similarity to the KS domain found in recently discovered HR type II PKS subfamily,

featuring a conserved cysteine residue (Cys146) crucial for the KS function[14] (Supplementary Figs. 13, 14A). As AmcG is a protein that lacks the functional motifs, it was difficult to predict the function before the protein was found to interact with AmcF. (Supplementary Table 1, Supplementary Fig. 13, Supplementary Fig. 14B). Thus, the function and structure of the AmcF−AmcG heterodimer are worthy of further attention.

The structure of the AmcF−AmcG heterodimer predicted by ColabFold[15,16] resembles those of typical KS−CLF complexes (PDB Code, 6KXD) with a cavity at the active center on the boundary between AmcF and AmcG where intermediates can be accommodated (Supplementary Fig. 15). The AmcF−AmcG dimeric structure validated by a SpeedPP pipeline[17,18] on the basis of AlphaFold2 shows a high-confidence model (pDockQ = 0.71) (Supplementary Fig. 16). As AmcF functions as a KS, AmcG might play a role in the cyclization of the $C_2$-elongated diketo substrate formed by the action of AmcF. Therefore, we propose that AmcG is referred to as a "cyclization factor (CYF)" to distinguish it from CLF. The facts that the structure of CYF lacks some of the secondary structures compared to CLF (Supplementary Fig. 17A−C) and that there is less protein sequence similarity between CLF and CYF even in the regions of common secondary structure (Supplementary Fig. 17D) suggest that CYF is a partner protein of KS belonging to a separate group from CLF. However, further biochemical analyses of CYF are required to clarify its characteristic function in the KS−CYF reaction.

Next, we added AmcE to the Reaction 1 mixture (designated Reaction 2 mixture) and incubated the Reaction 2 mixture at 30 °C for 1 h. The signal (11833.1 Da) corresponding to the mass of **2**-AmcB was significantly reduced, whereas a new signal (11789.1 Da) corresponding to the mass of **3**-AmcB was detected (Fig. 2A). The mass of **3**-AmcB was less than that of **2**-AmcB by 44 Da, corresponding to the mass of $CO_2$. The tandem MS-based phosphopantetetheinyl ejection assay also identified **3** as 5-oxocyclopent-1-ene-1-carboxylic acid (Fig. 2B). These results demonstrate that AmcE catalyzes the decarboxylation of **2**-AmcB to form **3**-AmcB. Thus, we successfully reconstituted the biosynthesis of **3**-AmcB using 5 recombinant enzymes, AmcB, AmcE, AmcF, AmcG, and AmcH, in vitro.

The function of AmcE, which was originally assigned as the DH of the PKS system, is also worth attention. Although AmcE is a member of the hot dog superfamily enzymes, which include thioesterase and DH, AmcE catalyzes decarboxylation. Among the hot dog superfamily enzymes, AmcE shows low similarity to 3-hydroxydecanoyl-ACP dehydratase FabA and 3-hydroxyacyl-ACP dehydratase FabZ for fatty acid synthesis (Supplementary Fig. 18A, B). The histidine residue near the substrate pocket in FabA and FabZ, which appears to be necessary for their catalysis[19], is highly conserved in AmcE (His66) and its homologs (Supplementary Fig. 18A, B), suggesting that His66 in AmcE may also be crucial for catalysis. On the other hand, AmcE differs from FabA and FabZ in the conserved residues near the substrate binding site (Supplementary Fig. 18), which may explain the difference between the DH activity of FabA and FabZ and the decarboxylase activity of AmcE. In recent years, the decarboxylase activity of several hot dog superfamily enzymes has been reported[20,21]. AmcE catalyzes elusive decarboxylation at a $sp^2$ carbon via an enzyme represented by orotidine-5′-monophosphate decarboxylase[22]. Thus, we propose the addition of AmcE to the hot dog superfamily as an enzyme exhibiting decarboxylase activity.

## Reconstitution of the biosynthesis of cispentacin from 3-AmcB

Heterologous production and bioinformatics analysis suggest that the putative aminotransferase AmcC and the thioesterase AmcD catalyze the reactions leading to cispentacin from **3**-AmcB (Supplementary Table 1).

AmcC is classified as a class-III transaminase, represented by pyridoxal 5′-phosphate (PLP)-dependent acetylornithine aminotransferase,

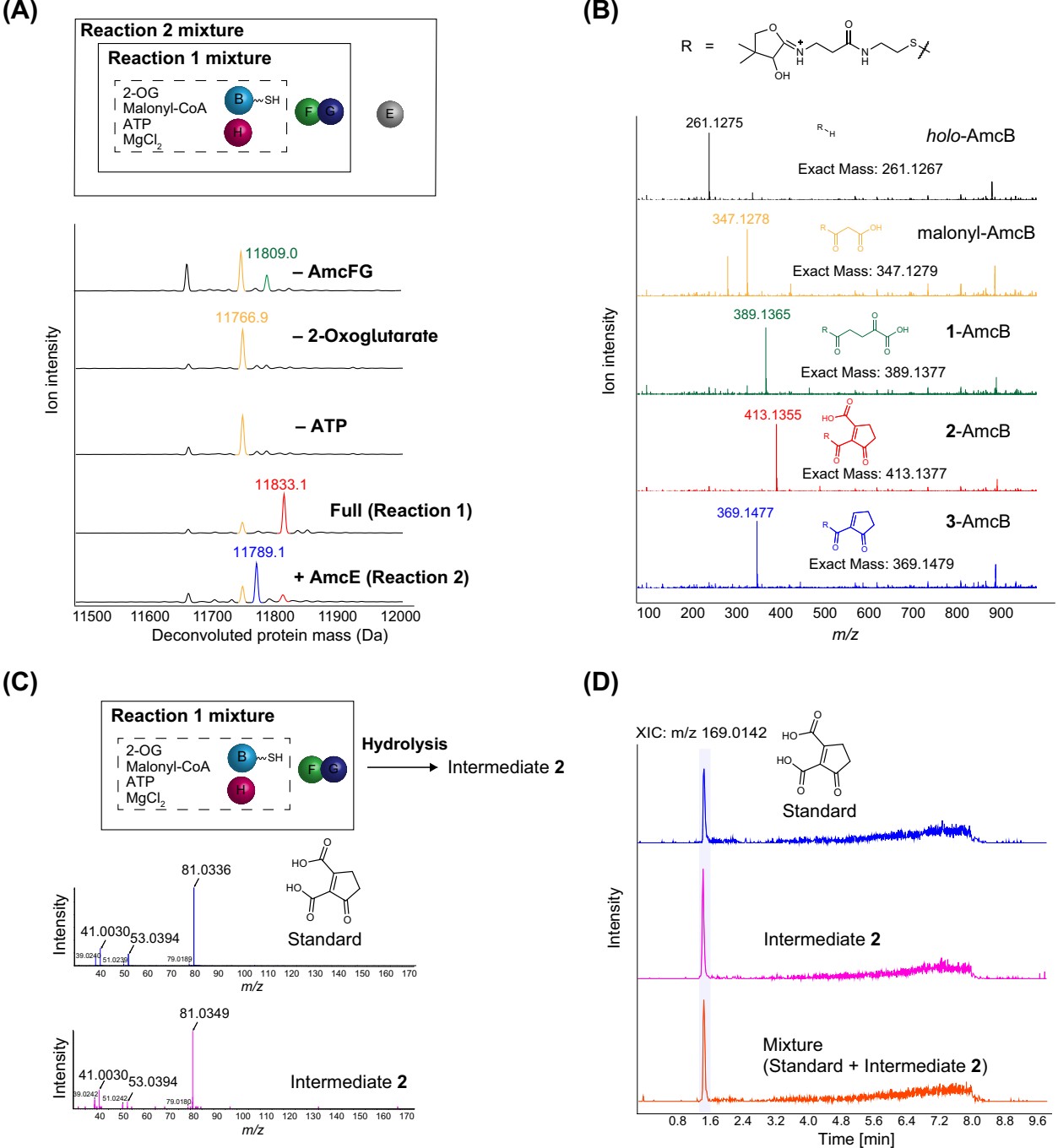

**Fig. 2 | Detection of intermediates bound to AmcB. A** Deconvoluted protein mass spectra for malonyl-AmcB (orange), **1**-AmcB (green), **2**-AmcB (red) and **3**-AmcB (blue). **B** MS/MS fragment of the 13+ charge state of *holo*-AmcB, malonyl-AmcB, **1**-AmcB, **2**-AmcB, and **3**-AmcB. Each PPant-eliminated ion described was clearly detected. **C 2**-AmcB was prepared in a Reaction 1 mixture and then digested from AmcB by alkaline hydrolysis. The MSMS spectrum from Intermediate **2** (magenta) matches well with that of the synthesized 3-oxocyclopent-1-ene-1,2-dicarboxylic acid standard (blue). **D** Comparison of extracted ion count (XIC) chromatograms at *m/z* 169.0142 of the standard (blue), Intermediate **2** (magenta), and a mixture of both (orange). A single peak of the mixture (orange) validates the identity of both compounds.

which transfers the α-amino group of L-glutamate to *N*-acetyl-L-glutamate 5-semialdehyde to yield 2-OG and *N*-acetylornithine. However, it was difficult to hypothesize both the amine donor and amine acceptor of AmcC. Thus, we decided to first perform the UV–vis spectra assay using potential amino group donors of AmcC. Usually, in PLP-dependent aminotransferase reactions, an internal aldimine (the enzyme-PLP complex) with $\lambda_{max} \approx 414$ nm is converted to pyridoxamine phosphate (PMP) with $\lambda_{max} \approx 323$ nm in an amine donor-dependent manner (first half-reaction of transamination), followed by the formation of the quinonoid intermediate (the PMP-an amine acceptor adduct) through dehydrative condensation between PMP and the amine acceptor. Therefore, we expected that we were able to identify the amine donor for AmcC simply by incubating AmcC with promising amine donors alone. Indeed, we detected PMP ($\lambda_{max} = 323$ nm) formation by AmcC in an L-ornithine- or L-lysine-dependent manner (Supplementary Fig. 19).

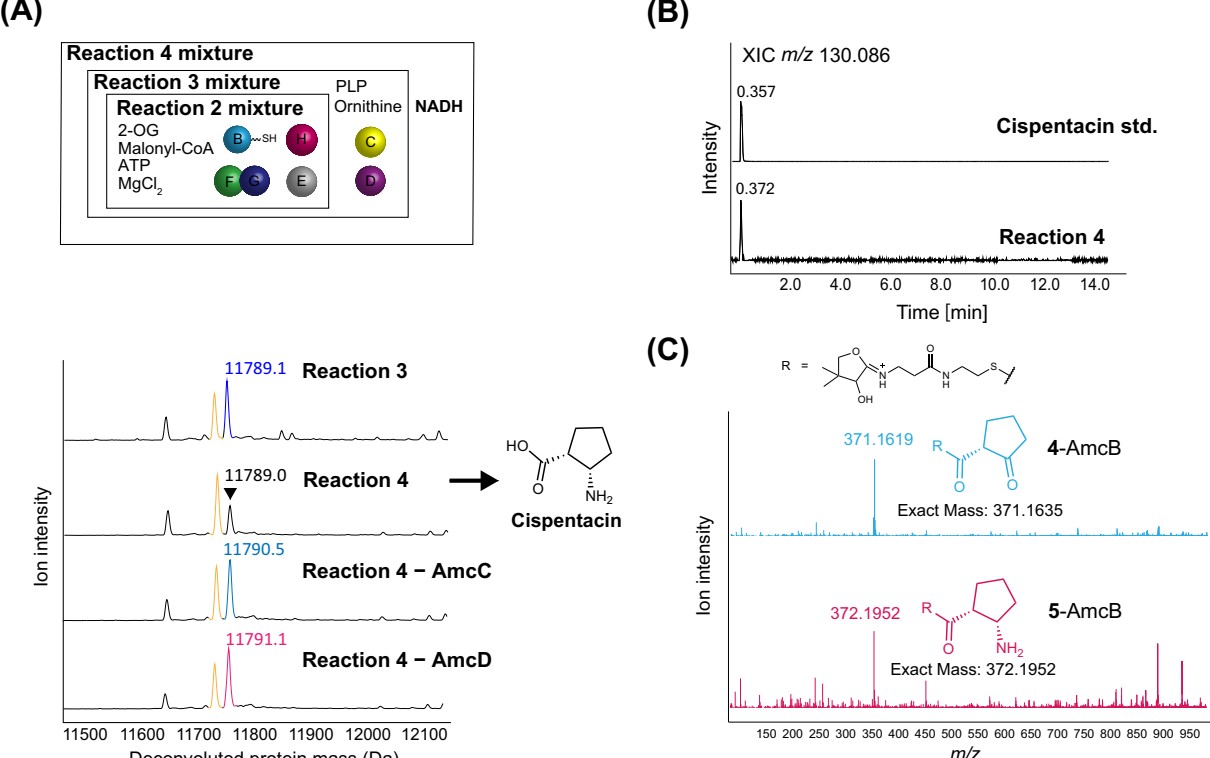

**Fig. 3 | Reconstruction of the pathway from 3-AmcB to cispentacin.**
**A** Deconvoluted protein mass spectra for each condition. **B** XIC chromatograms at
$m/z$ 130.08626 of the cispentacin standard and the Reaction 4 mixture. **C** MS/MS
fragments of 13 + charge state for **4**-AmcB and **5**-AmcB. The MS/MS fragment ($m/z$
371.1619) of **4**-amcB was detected in the condition without AmcC. The MS/MS
fragment ($m/z$ 372.1952) of **5**-AmcB was detected in the condition without AmcD.

Having identified the amine donor for AmcC, we next incubated
AmcC and AmcD with **3**-AmcB (prepared in Reaction 2 mixture), PLP as
the cofactor of AmcC, and L-ornithine as the amine donor for AmcC for
1 h (designated Reaction 3 mixture, Fig. 3A). However, no products
were detected. This result led us to assume that the reduction of the
cyclopentene ring of **3**-AmcB to the cyclopentane ring is necessary
prior to transamination by AmcC for the process leading to cispenta-
cin, and any reducing agent may thus be required for the process.
Interestingly, we found that only when NADH was added as a reducing
agent to the Reaction 3 mixture (designated Reaction 4 mixture) did
the deconvoluted protein mass signal for **3**-AmcB (11789 Da) sig-
nificantly decrease (Fig. 3A), clearly indicating the consumption of
**3**-AmcB. Subsequently, we confirmed the obvious production of cis-
pentacin in the Reaction 4 mixture (Figs. 3A, B, Supplementary Fig. 20).
Furthermore, a reduced intermediate, **4**-AmcB (**4**, 2-oxocyclopentan-1-
carboxylic acid) or **5**-AmcB (**5**, 2-aminocyclopentan-1-carboxylic acid),
was accumulated under conditions excluding AmcC or AmcD,
respectively, from the Reaction 4 mixture (Fig. 3C). In addition, the
AmcC-dependent conversion of **4**-AmcB to **5**-AmcB was detected when
L-ornithine or L-lysine was used as an amine donor (Supplementary
Fig. 21). These results indicate that AmcC is an unusual amino-
transferase that transfers the ω-amino group of ω-amino acids to the
keto group of C5 of **4**, which is bound to ACP. Furthermore, under the
condition that NADH was added to the Reaction 2 mixture, highly
efficient conversion of **3**-AmcB to **4**-AmcB was confirmed by both the
PPant ejection assay and the cysteamine-promoted cleavage assay[23]
(Supplementary Fig. 22). As the reduction did not occur without AmcE
or with NADPH as a reductant (Supplementary Fig. 22), an NADH-
dependent enzymatic reaction is essential for the reduction that gen-
erates **4**-AmcB from **3**-AmcB. However, no genes encoding enzymes
that might be involved in this reduction reaction were found in the *amc*
cluster.

To identify the enzyme responsible for the reduction, we
attempted to exclude the possibility of an unknown reaction
mechanism mediated by the Amc enzymes for the reduction reaction.
Thus, we decided to perform in vitro experiments using **3**-AmcB that
was further purified by gel filtration (Supplementary Fig. 23A). When
gel-filtered **3**-AmcB was incubated with each Amc enzyme and NADH at
30 °C for 1 h, conversion to **4**-AmcB was detected only when AmcB
solution was added (Supplementary Fig. 23B). Since AmcB, an ACP,
cannot catalyze NADH-dependent reduction alone, we hypothesized
that some *E. coli*-derived enzymes recruited to the AmcB ACP cata-
lyzed the reduction in the in vitro Reaction 4 (Fig. 3). The heterologous
production of cispentacin in *S. albus* G153 (Supplementary Fig. 4)
suggests that the universal FAS-related reductase (enoyl-ACP reduc-
tase, ER) that occurs across species catalyzes the reduction leading to
**4**-AmcB from **3**-AmcB. In addition, in prokaryotes such as *E. coli* and
*Streptomyces* strains, FAS forms a multienzyme system with KS, AT, KR,
DH, ER and ACP to increase the efficiency of fatty acid synthesis. Based
on mechanistic similarities, such as NADH selectivity and reduction of
C2−C3 double bonds in enoyl carboxylic acids, we hypothesized that
enoyl-ACP reductase (FabI) catalyzes the reduction of **3**-AmcB. To test
our hypothesis, we purified FabI (eFabI, derived from *E. coli*; sFabI,
derived from *S. albus* G153) as a soluble C-terminal His-tagged protein
and subsequently incubated each FabI with gel-filtered **3**-AmcB. LS-
MS/MS analyses clearly verified that eFabI and sFabI specifically cata-
lyze the reduction of **3**-AmcB to yield **4**-AmcB (Supplementary
Fig. 23C, D).

Finally, to determine the stereochemistry of **5**, we employed the
advanced Marfey's method with $N^α$-(5-fluoro-2,4-dinitrophenyl)-L-leu-
cinamide (L-FDLA) and $N^α$-(5-fluoro-2,4-dinitrophenyl)-D-leucinamide
(D-FDLA)[24–26]. The retention times in LC−MS and the MS/MS spectra of
the D- and L-FDLA-derivatized **5** intermediates matched well with those
of the cispentacin standard (Supplementary Fig. 24). This result

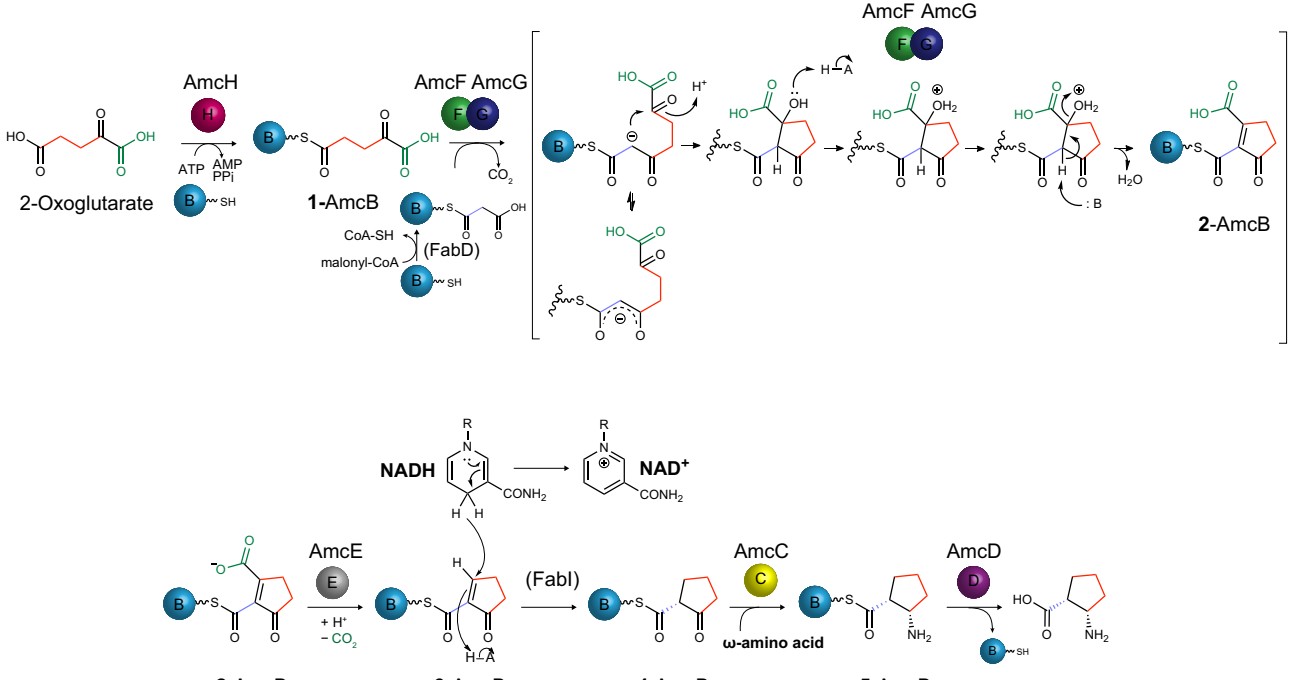

**Fig. 4 | Proposed biosynthetic pathway of cispentacin.** All reaction steps of the biosynthetic pathway from 2-OG to cispentacin were biochemically elucidated. Initially, 2-OG is modified to 2-oxoglutaryl-AmcB (**1**-AmcB) by AmcH adenylate-forming acyltransferase. **1**-AmcB then condenses with the elongation substrate malonyl-AmcB, followed by C2–C6 carbon bond formation and dehydration, which presumably involves residues on the enzyme acting as an acid and a base to form **2**-AmcB. This series of reactions is catalyzed by the AmcF–AmcG heterodimer.

**2**-AmcB is subsequently decarboxylated by the action of AmcE and converted to **3**-AmcB. Then, the FAS enzyme FabI, which is presumably recruited to AmcB, catalyzes the reduction of **3**-AmcB to form **4**-AmcB in the presence of NADH. Finally, **4**-AmcB is aminated to **5**-AmcB by AmcC aminotransferase in the presence of L-ornithine or L-lysine and then released from AmcB by AmcD-catalyzed hydrolysis to yield cispentacin.

demonstrates that the stereochemistry of **5** is identical to that of the cispentacin standard.

Taken together, the results indicate that the FAS enzyme FabI catalyzes the reduction of a C–C double bond of **3**-AmcB to form **4**-AmcB. Then, AmcC stereospecifically catalyzes transamination to a keto group at C2 of **4**-AmcB to form **5**-AmcB. AmcD-catalyzed hydrolysis of the thioester bond then occurs between **5** and the phosphopantetheine arm of AmcB in **5**-AmcB, resulting in the release of **5** (cispentacin) from AmcB (ACP). Consequently, we biochemically elucidated all steps of the biosynthetic pathway from 2-OG to cispentacin and established a total biosynthetic pathway for cispentacin (Fig. 4). Comparing the biochemical and structural characteristics of the AmcF–AmcG complex with those of the canonical type II KS-CLF complex led us to propose that AmcF–AmcG is the second subfamily of type II PKS, following the HR type II PKS.

### *amcG* homolog-containing BGCs are widespread across several bacterial phyla

Although the AmcF–AmcG (KS–CYF) heterodimer appears to be the canonical type II KS-CLF heterodimer in structural and biochemical features, AmcG (CYF) has no known functional motifs. AmcG is thus expected to be characteristic of the subfamily of type II PKS for the biosynthesis of skeletons such as cispentacin and CFA. Therefore, exploring unique BGCs containing *amcG* may lead to the discovery of natural products. In fact, bioinformatics analysis using the hmmer (http://hmmer.org/) and 2ndFind (http://biosyn.nih.go.jp/2ndfind/) tools revealed more than 150 bacterial species possessing the *amcG*-containing BGC (Fig. 5A and Supplementary Data 1). Of the *amcG*-containing BGCs, cispentacin-type BGCs (containing the 7 homolog genes of *amcB-H*) were conserved in a wide range of bacteria, including *Stigmatella* and *Corallococcus*, which are members of Myxobacteria; *Ignavibacteriae bacterium*, which is a member of a different phylum from actinomycetes and proteobacteria; and *B. cereus*, which is a cispentacin producer[1].

In addition, a BLASTP search using clustered non-redundant database (https://blast.ncbi.nlm.nih.gov/Blast.cgi) revealed that the cispentacin homologs are embedded in more than 100 diverse BGCs (Supplementary Data 2). Interestingly, almost all of these BGCs contain *amcB*, *amcF*, *amcG*, and *amcH* homologs (Supplementary Fig. 25A), which are indispensable for the biosynthetic pathway from 2-OG to 2-ACP. On the other hand, *amcC*, *amcD*, and *amcE* homologs exist only in some BGCs (Supplementary Fig. 25A). Based on the composition of the gene sets, the retrieved BGCs can be divided into at least three types: cispentacin-type (containing the seven homolog genes of *amcB-H*), CFA-type (containing the 5 homolog genes of *amcB*, *amcE*, *amcF*, *amcG*, and *amcH*), and minimal type (containing the four homolog genes of *amcB*, *amcF*, *amcG*, and *amcH*). The cispentacin-type BGCs presumably produce cispentacin via the biosynthetic pathway described above. The diversity of uncharacterized genes around the seven core genes in the cispentacin-type BGCs suggests the presence of unidentified natural products that contain the cispentacin moiety. (Fig. 5B). The CFA-type BGCs are expected to be responsible for producing unidentified natural products biosynthesized via the intermediates **3**-ACP or **4**-ACP (Fig. 5B). The minimal type of BGC may be responsible for producing unidentified natural products biosynthesized via the intermediate **2**-ACP (Fig. 5B). For example, the BLAST search mentioned above actually identified *Streptomyces scabies*, which has a CFA-type BGC and produces coronatine-like phytotoxins[27], and *Streptomyces* sp. B9173, which has the minimal type of BGC and produces FR900452[28,29]. In addition, several hypothetical BGCs for unknown natural products were found in bacteria, including *Lonsdalea quercina* and *Lonsdalea britannica*, which have CFA-type BGCs but lack

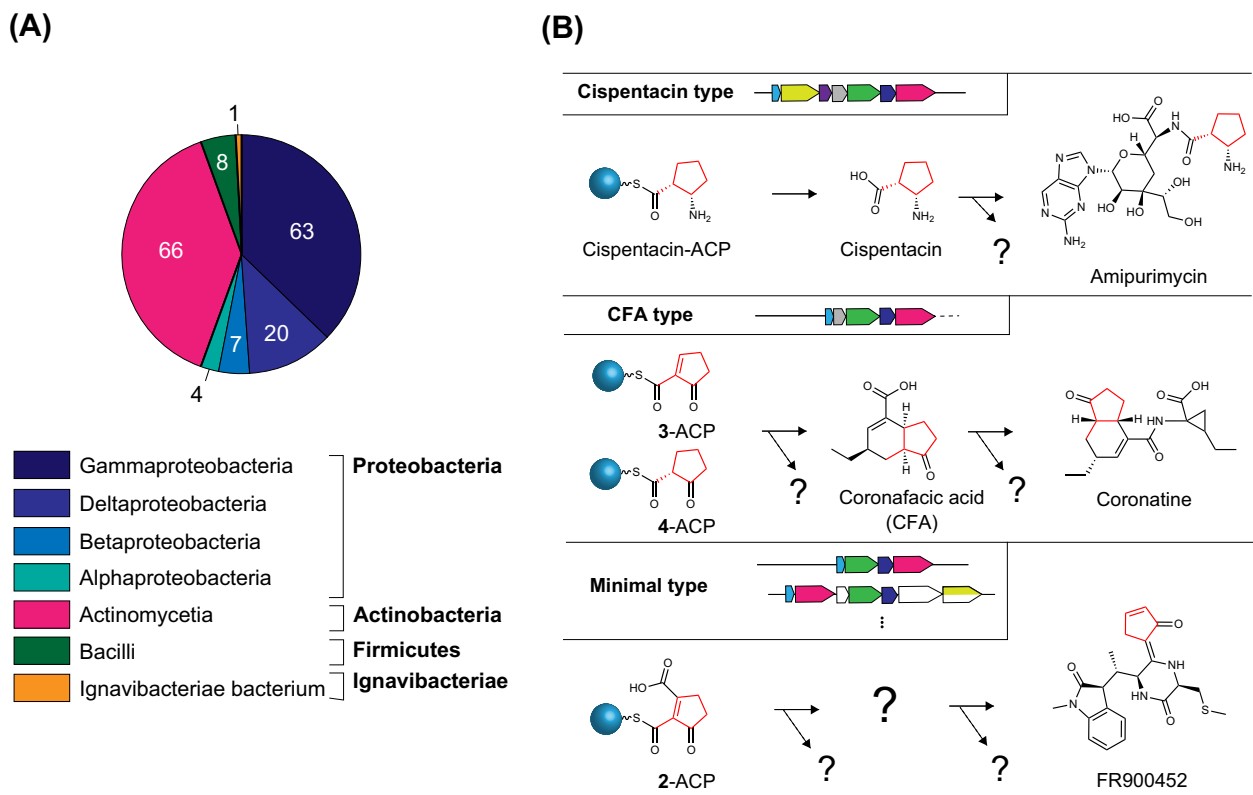

**Fig. 5 | *amcG* homolog-containing BGCs. A** Numbers of organisms harboring AmcG homologs. These organisms were extracted by the hmmer tool (phmmer search for UniProtKB). A list of these organisms is available in Supplementary Data 1. **B** Classification of BGCs and predicted divergence of the biosynthetic pathways.

one of the downstream type I PKS genes (*cfa7* homolog) and are predicted to produce a metabolite different from CFA (Supplementary Fig. 26A). Interestingly, *Agrobacterium tumefaciens*, *Paraburkholderia acidicola*, and *Burkholderia ubonensis*, which have the minimal type of BGC, including presumptive 9,11-endoperoxide prostaglandin H2 reductase, can be predicted to produce a prostaglandin-like metabolite not derived from arachidonic acid (Supplementary Fig. 26B).

Furthermore, a greater number of BGCs, including AmcH, responsible for early-step reactions can be predicted to be involved in the production of 2-OG-derived secondary metabolites. The diversity outside these core gene regions (Supplementary Fig. 25B) suggests the existence of a variety of unknown natural products produced by this characteristic biosynthetic machinery identified in this study.

## Discussion

In this study, we succeeded in the heterologous production of the nonproteinogenic amino acid cispentacin by expressing *amcB−amcH* in *S. albus*. In addition, based on the reconstitution of the in vitro production of cispentacin and the in vitro functional analyses of seven biosynthetic Amc proteins, we propose the biosynthetic pathway leading to cispentacin from 2-OG (Fig. 4). The stereochemistry of cispentacin biosynthesized in this study is identical to that of the cispentacin moiety, which is a partial structure of amipurimycin. In addition, the stereochemistry of the last AmcB-tethered intermediate **5** was experimentally verified to be identical to that of the end product cispentacin (Supplementary Fig. 24), demonstrating that aminotransferase AmcC introduces an amino group with (*S*)-stereochemistry into the C2 keto group of **4**. As described below, the stereochemistry at C1 of intermediate **4** was determined to be (*R*) by the FabI-catalyzed reaction mechanism. Thus, we propose a biosynthetic pathway for cispentacin, including the stereochemistry of all the intermediates (Fig. 4).

Our study provides three important insights into natural product biosynthesis. The first is a comprehensive understanding of the molecular basis for the production of cispentacin using biosynthetic genes. In recent years, cispentacin has also been used as a promising building block for nonstandard peptides[30]. Efficient production of cispentacin via biosynthesis may also contribute to the production of biologically active compounds. The second is the discovery of a biosynthetic strategy for natural product biosynthesis. A noteworthy feature of the cispentacin biosynthetic machinery is that noncanonical type II PKS-like enzymes, including ACP (AmcB) and KS (AmcF), produce a nonaromatic cyclopentane structure, in contrast to the canonical type II PKS enzymes that usually produce aromatic compounds. As the unique adenylation enzyme (AmcH) and "cyclization factor (CYF)" (AmcG) identified in this study are also required for these reactions for cispentacin biosynthesis, these enzymes may become recognized as a new subfamily of type II PKSs in the future. This study will also lead to unraveling the biosynthetic pathways of CFA and FR900452, both of which remain unknown.

The third interesting finding is the complementary and essential role of FabI, a FAS biosynthetic enzyme, in the cispentacin biosynthetic pathway. In this study, **3**-AmcB was successfully reduced to **4**-AmcB by FabI. This result is the second demonstration that FabI participates in secondary metabolism[31] and the demonstration that FabI can catalyze the reduction of the C2−C3 double bond of a natural cyclic substrate in addition to the linear substrate. FabI catalyzes the *syn* addition of 2H to C2−C3 double bonds of fatty acid intermediates (Supplementary Fig. 27)[32]. This is consistent with the (*R*)-C1 stereoisomerism of the end product cispentacin ((1*R*,2*S*)-2-aminocyclopentane-1-carboxylic acid) biosynthesized via FabI-catalyzed reduction. In addition, in the *amcG* homolog-containing BGC, no common genes that can substitute for the FabD and FabI reactions were found around the gene cluster. These findings support the

occurrence of crosstalk between the secondary biosynthetic machinery and the FAS biosynthetic machinery.

Furthermore, the search for homologs of AmcG, a key enzyme in the noncanonical type II PKS system, was also effective in finding relevant BGCs. The gene cassette harboring *amcG* is widespread across Proteobacteria, Actinomyces, Firmicutes and Ignavibacteriae. While some of them were predicted to construct known natural products, a number of BGCs are expected to be involved in the biosynthesis of unidentified natural products probably containing a five-membered carbon ring, such as cispentacin. As natural products such as amipurimycin, coronatine, and FR900452 show promising biological activities, the search for homologs of AmcG would contribute to the discovery of useful unknown natural products biosynthesized via the noncanonical type II PKS system identified in this study. It is also crucial to note that AmcG homologs can be found in plant pathogens such as *A. tumefaciens* and *L. quercina* (Supplementary Fig. 26).

## Methods

### Bacterial strains and associated chemicals
The bacterial strains in this study are listed in Supplementary Table 3. These strains were stored as mycelial glycerol stocks and kept at −80 °C.

All the chemicals, including the standard compound cispentacin, were purchased from commercial sources and used without further purification.

### Cloning
The vectors used in this study are listed in Supplementary Table 4. The BAC clone pKU503_ampr_P1-E8_R containing the amipurimycin biosynthetic gene (*amc*) cluster was used as template DNA for PCR. The oligonucleotides used in this study were purchased from Fasmac Co., Ltd. (Atsugi, Japan) and are listed in Supplementary Table 2. PCR was performed using KOD DNA polymerase (Toyobo, Osaka, Japan). The amplified PCR products were purified using agarose gel electrophoresis and a Gel Extraction kit (Qiagen, Tokyo, Japan). Purified PCR products were ligated into the pBluescript II SK(+) vector or pT7Blue vector (New England Biolabs, lpswich, MA) using DNA ligase (Toyobo, Osaka, Japan). *E. coli* DH5α was transformed with the resulting plasmids, which were then purified using a GenElute Plasmid Miniprep kit (Sigma–Aldrich) and subjected to DNA sequencing, which was outsourced (FASMAC, Atsugi, Japan).

### Detection and quantification of cispentacin
To confirm the production of cispentacin in *S. albus* G153/pSEamcB–H (see Supplementary Fig. 3 for details of the plasmid construct), a mycelial stock of *S. albus* G153/pSEamcB–H and the negative control strain *S. albus* G153/pSE101 were inoculated into 10 mL of TSB medium and incubated at 28 °C with shaking at 220 rpm for 2 days. A 100 µL portion of the seed culture was then separately inoculated in 10 mL of SEED medium (1.0% glucose, 4.0% soluble starch, 1.0% polypeptone, 0.45% pressed extract, 0.5% corn steep liquor, and 0.1% trace elements (pH 7.0)). After incubation at 28 °C with shaking at 220 rpm for 8 days, an equal volume of methanol was added to each culture, and the resulting mixtures were then centrifuged (10 min at 21,500 × *g*). Subsequently, the culture supernatant was derivatized by adding an equal volume of 1-fluoro-2,4-dinitrobenzene (DNFB) mixture (5% DNFB, 95% methanol, 100 mM NaOH), and the reaction mixture was then incubated at 60 °C for 2 h. After lyophilization of the reaction mixture, 400 µL water was added to the residue with subsequent extraction using 600 µL of ethyl acetate twice, followed by evaporation to dryness. The residues were dissolved in methanol and then centrifuged (10 min at 21,500 × *g*). Subsequently, the supernatants were subjected to LC-ESI analysis. The separation was confirmed by comparison with a standard cispentacin derivatized with DNFB (exact mass, 296.088 Da). MS/MS analysis also confirmed that the metabolite derivatized with

DNFB with *m/z* 296.088 [M + H]$^+$ could be detected in the production strains and showed the same fragmentation pattern as the derivatized standard cispentacin (Supplementary Fig. 4C). These derivatized samples were also subjected to X-LC analysis to quantify the titer of cispentacin by measuring its maximum absorbance at 360 nm (Supplementary Fig. 4A).

To quantify the cispentacin titer, the mycelial stock of the strains was inoculated in 10 mL of TSB medium and incubated at 28 °C with shaking at 220 rpm for 2 days. A 2 mL portion of the seed culture was then inoculated in 100 mL of production medium (2% malt extract, 4% sucrose, 3% soy flour, and 0.6% NaCl). After incubation at 28 °C with shaking at 220 ppm for an additional 12 days, an equal volume of methanol was added to the cells and then centrifuged (10 min at 21,500 × *g*). Then, 1 mM L-phenylalanine (Peptide Institute, Osaka, Japan) was added to each sample as an internal control. The supernatant was derivatized by adding an equal volume of DNFB mixture (5% DNFB, 95% methanol, and 100 mM NaOH) and then incubated at 60 °C for 2 h. After lyophilization of the reaction mixture, 400 µL water was added to the residue with subsequent extraction using 600 µL of ethyl acetate twice, followed by evaporation to dryness. The samples were dissolved in methanol and then centrifuged (10 min at 21,500 × *g*). Subsequently, the supernatants were subjected to X-LC analysis. The cispentacin titer was accurately quantified by standardizing to the height of the peak derived from phenylalanine.

### Purification of recombinant proteins from *E. coli* BL21(DE3)
The pHis8 vector[33] and pET26b(+) vector were used for construction of the plasmids listed in Table S4. Each gene (*amcB, amcC, amcD, amcE, sfabI* and *efabI*) was amplified using the specific set of primers shown in Table S2. Amplified DNA fragments for each full-length gene (*amcC, amcD,* and *amcE*) were purified and ligated into the pHIs8 vector. other genes (*amcB, sfabI* and *efabI*) were introduced into the pET26b(+) vector.

After introducing each plasmid into *E. coli* BL21(DE3), each transformant was cultivated in 200 mL TB medium (tryptone 1.2%, yeast extract 2.4%, glycerol 0.56%, and K$_2$HPO$_4$ 0.23% supplemented with the corresponding antibiotic) at 37 °C for 2 h. After cooling the culture on ice for 10 min, β-D-thiogalactopyranoside (IPTG) was added to the culture at a final concentration of 100 µM to induce gene expression, and the culture was further incubated at 18 °C for 12 h. The cells were harvested by centrifugation and washed with 20 mL wash buffer (20 mM Tris-HCl (pH 8.0), 300 mM NaCl, 20 mM imidazole-HCl (pH 8.0), and 20% glycerol), followed by centrifugation. The washed pellet was resuspended in wash buffer again and lysed by sonication on ice. Cell debris was removed by centrifugation at 4 °C (30,000 × *g*, 20 min). The supernatant was added to an Econo-Pac® chromatography column (Bio-Rad, Hercules, CA), and the resins were washed with 30 mL of wash buffer. Then, proteins were eluted in 10 mL of elution buffer (20 mM Tris-HCl (pH 8.0), 300 mM NaCl, 250 mM imidazole-HCl (pH 8.0), and 20% glycerol). Each purified protein sample was examined by SDS–PAGE. Finally, each purified protein sample was concentrated using a Vivaspin® 20 ultrafiltration unit (Sartorius) followed by the addition of storage buffer (20 mM Tris-HCl (pH 8.0), 1 mM DTT, and 20% glycerol (pH 8.0)), and the resulting solutions were then aliquoted and stored at −80 °C.

### Purification of recombinant proteins from *S. albus* G153
The *Streptomyces–E. coli* shuttle vector pSE101[34] was used for the expression and purification of recombinant proteins in the *S. albus* G153 strain. The plasmid pSE101-*amcF/amcG* was constructed by subcloning the sequenced *amcF/amcG* DNA fragment into the *Hind*III and *Xba*I sites downstream of *lacP* on vector pSE101. pSE101-*amcF/amcG* was introduced into *S. albus* G153 using the polyethylene glycol-mediated protoplast method[35]. Each transformant was precultivated in 10 mL of TSB medium with 30 µg/mL thiostrepton (Tsr) at 30 °C and

300 rpm for 2 days. A 200 μL portion of the seed culture was inoculated into 100 mL of YEME medium (10.3 g sucrose, 0.3 g yeast extract, 0.5 g peptone, 1.0 g glucose, 0.3 g malt extract, separately autoclaved 0.5 mL 1 M MgCl₂ 6H₂O) containing 30 μg/mL Tsr and incubated at 28 °C for 3 days. The procedures for protein purification were the same as those for *E. coli* BL21(DE3).

## Gel filtration of AmcF–AmcG

AmcF and AmcG purified for in vitro reconstitution are expected to form a heterodimeric structure. The enzymes were applied to a Superdex 200 Increase 10/300 GL gel filtration column (Cytiva) equivalent to a buffer containing 20 mM Tris-HCl (pH 8.0) and 150 mM NaCl. The samples were eluted using the same buffer. The molecular weight of the enzymes in the solution was calculated based on a calibration curve prepared using the Gel Filtration Calibration Kit HMW (Cytiva) and compared with the theoretical molecular weights of the AmcF–AmcG heterodimer. The fractionated proteins were confirmed to be AmcF–AmcG by SDS–PAGE.

## In vitro reconstitution

**Reaction mixture to analyze the substrate specificity of AmcH.** The reaction was performed in a total volume of 50 mL of buffer comprising 50 mM Tris-HCl (pH 8.0) and 25 mM MgCl₂, and the mixture also contained 200 μM AmcB (ACP), 0.2 μM AmcH (AT) and necessary substrates (1 mM starter substrate and 5 mM ATP) and was incubated at 30 °C for 1 h. After incubation, each reaction mixture was analyzed using the following procedure.

**Reaction 1 mixture.** In vitro production of 2-ACP in the reaction mixture was assessed in a total volume of 50 mL of buffer comprising 50 mM Tris-HCl (pH 8.0) and 25 mM MgCl₂, and the mixture contained 200 μM AmcB (ACP), 0.2 μM AmcH (AT), 8 μM AmcF–AmcG (KS–CYF), and necessary substrates (1 mM malonyl-CoA, 1 mM 2-OG, and 5 mM ATP) and was incubated at 30 °C for 1 h. After incubation, each reaction mixture was analyzed using the following procedure.

**Reaction 2 mixture.** The reaction for in vitro production of **3**-ACP was performed in a total volume of 50 μL of buffer comprising 50 mM Tris-HCl (pH 8.0) and 25 mM MgCl₂, and the mixture also contained 200 μM amcB (ACP), 0.2 μM AmcH (AT), 8 μM AmcF–AmcG (KS-CYF), 8 μM AmcE (DH-like enzyme), and necessary substrates (1 mM malonyl-CoA, 1 mM 2-OG, and 5 mM ATP) and was incubated at 30 °C for 1 h. After incubation, each reaction mixture was analyzed using the following procedure.

**Reaction 3 mixture.** The reaction for in vitro production of cispentacin was performed in a total volume of 50 μL of buffer comprising 50 mM Tris-HCl (pH 8.0) and 25 mM MgCl₂, and the mixture also contained 200 μM AmcB (ACP), 0.2 μM AmcH (AT), 8 μM AmcF–AmcG (KS-CYF), 8 μM AmcE (DH-like enzyme), and necessary substrates (1 mM malonyl-CoA, 1 mM 2-OG, and 5 mM ATP) and was incubated at 30 °C for 1 h for production of 3-ACP. The mixture containing 8 μM AmcC and 8 μM AmcD, with a necessary substrate (10 mM ornithine) and cofactor (1 mM pyridoxal phosphate) added, was then incubated at 30 °C for 1 or 3 h. After incubation, each reaction mixture was analyzed using the following procedure.

**Reaction 4 mixture.** The reaction for in vitro production of cispentacin was performed in a total volume of 50 μL of buffer comprising 50 mM Tris-HCl (pH 8.0) and 25 mM MgCl₂, and the mixture also contained 200 μM AmcB (ACP), 0.2 μM AmcH (AT), 8 μM AmcF–AmcG (KS-CYF), 8 μM AmcE (DH-like enzyme), and necessary substrates (1 mM malonyl-CoA, 1 mM 2-OG, and 5 mM ATP) and was incubated at 30 °C for 1 h for production of 3-ACP. The mixture containing 8 μM

AmcC and 8 μM AmcD with necessary substrates (10 mM ornithine and 1 mM NADH) and co-factor (1 mM pyridoxal phosphate) added was then incubated at 30 °C for 1 or 3 h. After incubation, each reaction mixture was analyzed using the following procedure.

## Validation of the reduction of 3-AmcB to 4-AmcB

To identify the enzyme responsible for catalyzing the reduction of **3**-AmcB to **4**-AmcB using NADH as a reductant, we conducted the following steps.

**Gel filtration purification of 3-AmcB.** **3**-AmcB (prepared from the Reaction 2 mixture) was subjected to gel filtration using a Superdex 200 Increase 10/300 GL column (Cytiva) with a buffer containing 20 mM Tris-HCl (pH 8.0) and 150 mM NaCl. The resulting fractions were analyzed using SDS–PAGE and LC–HRMS to confirm the presence of AmcB. Subsequently, the fractions were concentrated using a Vivaspin® 20 ultrafiltration unit (Sartorius).

**Reduction reaction.** The reduction reaction was performed in a total volume of 10 μL of buffer, comprising 20 mM Tris-HCl (pH 8.0) and 150 mM NaCl. The reaction mixture also contained 200 μM gel-filtered AmcB, 8 μM of each enzyme (AmcB, AmcC, AmcE, AmcF-AmcG, AmcH, sFabI, or eFabI), and 1 mM NADH. The incubation was carried out at 30 °C for 1 h, and then each reaction mixture was analyzed using the PPANT ejection assay and LC–HRMS methods described below. We also tested the conditions for active AmcB, eFabI, and sFabI by subjecting the enzyme solutions to heat treatment at 98 °C for 10 min. This was performed to verify the effect of heat inactivation on the enzymatic activities.

## In vitro ACP assay

For analysis of ACP, after incubation of the reaction mixture, the ACPs were concentrated using Amicon Ultra 10 K devices (Merck Millipore). The pellet was redissolved in 50 μL of 50% methanol in water and centrifuged at 4 °C (21,500 × *g*, 10 min). Subsequently, the supernatants were subjected to LC–HRMS analysis as described later. LC–HRMS data of intact protein were analyzed using a PeakView software (SCIEX) protein deconvolution (Reconstruct protein) by the SCIEX algorithm.

## Tandem MS-based phosphopantetheinyl (PPANT) ejection assay

A tandem MS-based PPANT ejection assay was performed in two ways. The first method involves digestion of ACP samples using the peptidase thermolysin (Promega, Madison, US) to analyze mass derived from small peptide molecules. A 10 μL aliquot of the sample obtained in the procedure of the "In vitro ACP assay" described above was combined with 15 μL of thermolysin solution (166 μM thermolysin, 50 mM Tris-HCl, 0.5 mM CaCl₂) and digested at 50 °C for 1 h. After incubation, the reaction in each mixture was quenched using 225 μL of 50% methanol and then centrifuged at 4 °C (21,500 × *g*, 10 min). Subsequently, the supernatants were subjected to LC–HRMS analysis as described later. Detection of the PPANT-eliminated ion produced from the AmcB-bound-intermediate was conducted in IDA explore mode by searching for the precursor ion with a specific fragment ion at *m/z* 261.1267[12]. The PPANT-eliminated ions produced from **2**-AmcB and **3**-AmcB were detected, and the MS/MS fragments provided information on the structure of the intermediate (Supplementary Fig. 12). The second method was used to directly identify the PPANT-eliminated ions from the multiply charged ions of intact AmcB and AmcB-bound intermediates. We detected PPANT-eliminated ions from +13 charge states of *holo*-AmcB, malonyl-AmcB, **1**-AmcB, **2**-AmcB, **3**-AmcB, **4**-AmcB and **5**-AmcB through LC–HRMS analysis of the sample obtained in the procedure of the "In vitro ACP assay" described above.

## Cysteamine-promoted cleavage assay

Cysteamine hydrochloride was added to the reaction mixture at a final concentration of 0.2 M. After incubation at 30 °C for 45 min, the solution was treated by the addition of 100 μL acetonitrile followed by mixing. The resulting solution was then centrifuged at 4 °C ($21,500 \times g$, 10 min), and the supernatant was subjected to LC−HRMS analysis.

## LC−HRMS analysis

Metabolites from crude extraction and products from in vitro assays were analyzed by LC−ESI−HRMS using a SCIEX triple TOF 5600 system equipped with an Ultra-Fast Liquid Chromatography (UFLC) Nexera system (Shimazu; Kyoto, Japan). The UFLC system was equipped with a CAPCELLPAK C18 IF column (2.0 mm × 50 mm; Shiseido, Tokyo, Japan) and eluted at a flow rate of 0.4 mL/min. The Proteonavi C4 column (2.0 mm × 50 mm; Shiseido, Tokyo, Japan) was used to analyze protein samples and conduct cysteamine-promoted cleavage assay. The composition of the mobile phases was as follows: A, water + 0.1% formic acid; B, acetonitrile + 0.1% formic acid. The program was 10% B for 1 min, a linear gradient of 10−90% B for 4 min, 90% B for 1 min, and 10% B for 5 min; flow rate, 0.4 mL min$^{-1}$. MS and MS/MS parameters were as follows: ESI positive ion mode; acquisition mass range, $m/z$ 50−1500; TOF accumulation time, 0.1 S; MS/MS accumulation time, 0.1 s; collision energy, 30 eV; ion source gas 1, 50 psi; ion source gas 2, 50 psi; curtain gas 25 psi; ion spray voltage floating, 5500 v; temperature 550 °C. MS data were analyzed using a PeakView software (SCIEX).

## HPLC analysis for AmcH activity

To detect AMP, a byproduct of 2-oxoglutarate-ACP-ligase, the AmcH reaction was performed in a total volume of 50 μL of buffer comprising 50 mM Tris-HCl (pH 8.0) and 25 mM $MgCl_2$, and the mixture also contained 500 μM amcB (ACP), 1 μM amcH, and necessary substrates (1 mM 2-OG and 2 mM ATP) and was incubated at 30 °C for 1-4 h. After incubation, each reaction mixture was quenched by adding an equal volume of methanol. The resulting solution was then diluted 10-fold using 50% methanol and centrifuged at 4 °C ($21,500 \times g$, 10 min). Then, the supernatants were analyzed on an HPLC system equipped with an MD-2010 Plus Photodiode array (JASCO, Tokyo, Japan) and a CAPCELL PAK C18 Mg II column (4.6 × 250 mm; OSAKA SODA, OSAKA, Japan) under the following conditions: mobile phase A, water + 5 mM dibutylammonium acetate; mobile phase B, MeOH to 95% An over 20 min, 5% A for 10 min; flow rate, 1.0 mL min$^{-1}$.

## X-LC analysis

Samples were analyzed on an X-LC system equipped with an X-LC 3110MD photodiode array (JASCO, Tokyo, Japan) and a CAPCELL PAK C18 IF column (2.0 mm × 50 mm; OSAKA SODA, OSAKA, Japan) under the following conditions: mobile phase A, water + 0.1% formic acid; mobile phase B: acetonitrile + 0.1% formic acid [a linear gradient of 10−90% B for 10 min, 90% B for 2.5 min, and 10% B for 2.5 min; flow rate, 0.4 mL min$^{-1}$] or [a linear gradient of 10−50% B for 10 min, 90% B for 2.5 min, and 10% B for 2.5 min; flow rate, 0.4 mL min$^{-1}$]. The former conditions were used to quantify cispentacin, and the latter were used for the other analyses. Absorbance was scanned in the wavelength range 200−650 nm. The titer of cispentacin was calculated from the peak area of the DNFB-derivatized cispentacin at 360 nm.

## UV−vis analysis of AmcC

UV−vis analysis was performed with a UV-1900i (Shimazu). The AmcC reaction for UV−vis analysis was performed in a mixture containing 50 mM Tris-HCl (pH 8.0), 100 μM AmcC, and 1 μM to 10 mM amine donor (L-ornithine, L-glutamate, L-lysine, L-glutamine, and L-arginine) in a total volume of 500 μL. Measurements were taken at room temperature and immediately after the addition of amine donor.

## Structure prediction and scoring

ColabFold ver1.5.2[16] was used to predict the structures of the AmcF−AmcG complex and the AmcE dimer. All structures were predicted as multimer using default parameters. We used the highest scored model of the five outputs to perform further analysis using PyMOL software (Schrodinger LLC.). To validate the reliability of the complex structure interactions, the SpeedPPI pipeline based on FoldDock[17,18] and AlphaFold2 was employed. The All-vs-all mode with default parameters was used for the analysis. The structure outputs were obtained as High_confidence_preds, and the pdockQ values were used as ppis filtered.

## Synthesis of 3-oxocyclopent-1-ene-1,2-dicarboxylic acid

3-Oxocyclopent-1-ene-1,2-dicarboxylic acid (**2**) was synthesized following methods described previously[36,37]. Initially, a solution of *tert*-butyl 2-(triphenylphosphoranylidene)acetate (15 g, 40 mmol) in dry benzene (100 mL) at 0 °C was slowly added dropwise to a solution of 3-chloropropionyl chloride (2.54 g, 20 mmol) in dry benzene (16 mL). After the acyl chloride was completely added, the reaction mixture was stirred at room temperature for 30 min. The salt was removed by filtration after 120 mL of dry ether was added. The resulting filtrate was concentrated under reduced pressure, yielding 8.38 g of crude 1,1-dimethylethyl 5-chloro-3-oxo-2-(triphenylphosphoranylidene) pentanoate. Next, crude 1,1-dimethylethyl 5-chloro-3-oxo-2-(triphenylphosphoranylidene)pentanoate (3.90 g) in THF (94 mL) and water (24 mL) was treated with oxone® (8.60 g, 14.0 mmol) and vigorously stirred for 4 h. After the reaction, the mixture was diluted with water, and the layers were separated. The aqueous layer was further extracted with EtOAc. The combined organic extracts were dried over $Na_2SO_4$, filtered, and concentrated under reduced pressure, affording crude as a yellow oil. This crude was filtered through silica to remove byproducts and concentrated in vacuo to yield a yellow oil (1.7 g). This yellow oil was then diluted in THF (32 mL) and treated with saturated aqueous $NaHCO_3$ (43 mL). The resulting mixture was vigorously stirred for 4 h before being diluted with water. The layers were separated, and the aqueous layer was subjected to extraction with EtOAc. The combined organic extracts were dried over $Na_2SO_4$, filtered, and concentrated under reduced pressure. The residue was purified by chromatography using $SiO_2$ (hexane/$Et_2O$ = 3:1) to 2,3-dioxopent-4-enoic acid *tert*-butyl ester (654 mg, 39% from 1,1-dimethylethyl 5-chloro-3-oxo-2-(triphenylphosphoranylidene)pentanoate) as a yellow solid. Subsequently, a solution of *tert*-butyl 2-(triphenylphosphoranylidene)acetate (376 mg, 1 mmol) in ethyl acetate was added dropwise over 30 min to 2,3-dioxopent-4-enoic acid *tert*-butyl ester (202 mg, 1 mmol) in ethyl acetate (80 mL). After the reaction mixture was stirred for 35 min at 0 °C, it was concentrated under reduced pressure. Chromatography was performed on the resulting residue using $SiO_2$ and a mixture of 20% ether/hexane, providing 2,3-di(carbo-*tert*-butoxy)−2-cyclopentenone as a white solid (74%, 210 mg). The NMR spectra obtained for 2,3-di(carbo-*tert*-butoxy)−2-cyclopentenone are shown in Supplementary Fig. 28A–D. To obtain 3-oxocyclopent-1-ene-1,2-dicarboxylic acid, 2,3-di(carbo-*tert*-butoxy)−2-cyclopentenone (105 mg) was diluted in 0.4 mL of trifluoroacetic acid (TFA)/dichloromethane (1:1) and stirred for 1.5 h at 0 °C. TFA in the reaction mixture was eliminated by adding toluene and subjecting the mixture to azeotropic distillation. Purification using preparative HPLC with MeOH/$H_2O$ (10:90, v/v, 8 mL/min) resulted in the isolation of 3-oxocyclopent-1-ene-1,2-dicarboxylic acid (18 mg). The NMR spectra obtained for 3-oxocyclopent-1-ene-1,2-dicarboxylic acid are shown in Supplementary Fig. 29A, B. Importantly, this compound readily undergoes significant decomposition in organic solvents; thus, only short-duration NMR analysis can be performed when the compound is dissolved in deuterated DMSO-$d_6$.

## Analysis of 2-AmcB hydrolysates

To verify the structure of intermediate **2**, **2**-AmcB produced in the Reaction 1 mixture was hydrolyzed with 300 mM KOH at 65 °C for 30 min. After acidification with HCl (pH 2–3), the reaction mixture was filtered using Amicon Ultra 3 K devices (Merck Millipore) to remove protein. The permeate fraction was neutralized with KOH (pH 7) for LC–MS analysis. For the LC–MS analysis, synthesized 3-oxocyclopent-1-ene-1,2-dicarboxylic acid was diluted with water and used as an authentic reference.

Synthesized 3-oxocyclopent-1-ene-1,2-dicarboxylic acid and the hydrolysates from the Reaction 1 mixture were analyzed by LC-ESI–HRMS using a SCIEX triple TOF X500R system equipped with an UFLC Nexera system (Shimazu; Kyoto, Japan). The UFLC system was equipped with a CAPCELLPAK C18 IF column (2.0 mm × 50 mm; Shiseido, Tokyo, Japan) and eluted at a flow rate of 0.4 mL/min. The composition of the mobile phases was as follows: A, water + 0.1% formic acid; B, acetonitrile + 0.1% formic acid. The program was a linear gradient of 1–90% B for 5 min, 90% B for 2.5 min, and 1% B for 2.5 min; flow rate, 0.4 mL min$^{-1}$. MS and MS/MS parameters were as follows: ESI negative ion mode; acquisition mass range, $m/z$ 50-500; TOF accumulation time, 0.1 S; MS/MS accumulation time, 0.1 s; collision energy, −10 eV; ion source gas 1, 60 psi; ion source gas 2, 60 psi; curtain gas 30 psi; ion spray voltage floating, −4500 v; temperature 350 °C. MS data were analyzed using an Analyst software (SCIEX). The HRMS spectra obtained for 3-oxocyclopent-1-ene-1,2-dicarboxylic acid and intermediate **2** are shown in Supplementary Fig. 30.

## Analysis of 5-AmcB hydrolysate using the Marfey's method

To determine the stereochemistry of intermediate **5**, **5**-AmcB hydrolysates were analyzed by the advanced Marfey's method[24–26]. **5**-AmcB produced in the Reaction 5 mixture (described later) was hydrolyzed with 300 mM KOH at 65 °C for 30 min. After acidification with HCl (pH 2–3), the reaction mixture was filtered using Amicon Ultra 3 K devices (Merck Millipore) to remove protein. The permeate fraction was neutralized with KOH (pH 7) and filled up to 50uL by mq. To the neutralized solution, 20 µL of 1 M sodium bicarbonate and then 100 µL of 1% $N^{\alpha}$-(5-fluoro-2,4-dinitrophenyl)-L-leucinamide (L-FDLA) or $N^{\alpha}$-(5-fluoro-2,4-dinitrophenyl)-D-leucinamide (D-FDLA) in acetone were added. The solution was vortexed and incubated at 37 °C for 60 min. After the reaction was quenched by the addition of 20 µL of 1 N HCl, the reaction mixture was diluted with 200 µL of acetonitrile and then analyzed by LC–ESI–HRMS via a SCIEX triple TOF X500R system equipped with an UFLC Nexera system (Shimazu; Kyoto, Japan). The UFLC system was equipped with a CAPCELLPAK C18 IF column (2.0 mm × 50 mm; Shiseido, Tokyo, Japan) and eluted at a flow rate of 0.4 mL/min. The composition of the mobile phases was as follows: A, water + 0.1% formic acid; B, acetonitrile + 0.1% formic acid. The program was a linear gradient of 10–50% B for 4 min, 50–100% for 16 min, 100% B for 3 min, and 10% B for 2 min; the flow rate was 0.4 mL min$^{-1}$. The MS and MS/MS parameters were as follows: ESI negative ion mode; acquisition mass range, $m/z$ 50-500; TOF accumulation time, 0.1 s; MS/MS accumulation time, 0.1 s; collision energy, −10 eV; ion source gas 1, 60 psi; ion source gas 2, 60 psi; curtain gas 30 psi; ion spray voltage floating, −4500 v; temperature 350 °C. MS/MS data were analyzed using an Analyst software (SCIEX).

**Reaction 5 mixture.** The reaction for the in vitro production of intermediate **5** was performed in a total volume of 50 µL of buffer comprising 50 mM Tris-HCl (pH 8.0) and 25 mM MgCl$_2$ at 30 °C for 2 h. The mixture also contained 200 µM AmcB (ACP), 0.2 µM AmcH (AT), 8 µM AmcF–AmcG (KS–CYF), 8 µM AmcE (DH-like enzyme), 8 µM AmcC, 8 µM eFabI and necessary substrates (1 mM malonyl-CoA, 1 mM 2-OG, and 5 mM ATP, 1 mM ornithine and 1 mM NADH) and co-factor (1 mM pyridoxal phosphate).

## Reporting summary

Further information on research design is available in the Nature Portfolio Reporting Summary linked to this article.

## Data availability

The lists of organisms possessing the *amcG*-containing BGC is provided in Supplementary Data 1 and Supplementary Data 2. Previously published structures cited in this study can be accessed using PDB accession numbers 6kxd, 6kxf, 6qsp, 1mka, and 6n3p. The nucleotide sequence of the amc cluster is deposited in the DDBJ/EMBL/GenBank nucleotide sequence database under the accession number LC389220. Other data are available from the corresponding authors upon request. Source data are provided with this paper.

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

## Acknowledgements

This work is supported by grants from JSPS KAKENHI (16H06453 and 22H05120) to T.K.

## Author contributions

Research planning and supervision were M.N., and T.K.; experiments except chemical synthesis were performed by G.H. and T.U. Synthesis of 3-oxocyclopent-1-ene-1,2-dicarboxylic acid was performed by Y.O., K.N., T.S., and G.H.; and the manuscript was written by G.H., T.S., and T.K.

## Competing interests

The authors declare no competing interests.
