## [Peer review file · Nature Communications]

Reviewer #1 (Remarks to the Author):

The manuscript "Unprecedented Type II Polyketide Synthase-like Enzymes that Expand Natural Product Diversity" by Hibi, et al., describes studies into the biosynthesis of cispentacin from amipurimycin producing streptomycetes. Fascinatingly, it appears that this 5-membered cyclic β -amino acid is prepared from α -ketoglutarate from action of a collection of type II PKS enzymes, including an ACP, AT, KS, and CLF-like stand-alone domains. This work has the potential to dramatically revise our definitions of type II PKS enzymes, which had already been recently revised through the elucidation of polyene PKS pathways. Despite the uniqueness of these studies, the manuscript contains significant conceptual and experimental deficiencies that must be corrected and/or completed before this study can be sufficient for publication. The following major items should be addressed before further determination:

1. Page 4. Line 12. "It was also confirmed that the resulting holo-AmcB was indeed converted to malonyl-AmcB (11767 Da) in the presence of malonyl-CoA in vitro (Figure 14 S8)." This claimed result is an artifact of acyl transfer that occurs in in vitro experimentation that has now been very well documented. This spontaneous acylation cannot serve as the physiological malonyltransfer mechanism, as it is likely that other CoA species (such as acetyl-CoA) will predominate in vivo.
2. Page 4, line 23. "AmcH is the first adenylate-forming acyltransferase (AT) that forms 2-oxoglutaryl-ACP." The use of acyltransferase (AT) nomenclature is incorrect. ATs are CoA dependent enzymes, whereas were the enzyme is shown to be ATP dependent. This classifies it as an adenylating enzyme (acyl-AMP forming), abbreviated as A.
3. Creating the new nomenclature "cyclization factor (CYF)" is a new concept and should be evaluated by all reviewers. If the proposed mechanism is fully demonstrated, I approve of the new name.
4. Page 7 Line 19-34. Again, this study is dependent upon a spontaneous reaction, here reduction of 3-AmcB to 4-AmcB by NADH. While such an outcome may be visualized for in vitro experimentation, it is inconceivable that this could be the physiological mechanism. The proper enzyme that catalyzes this reaction must be identified. Could it be FabI, or perhaps there is an adjacent reductase in the operon? The authors must conduct this due diligence before this study can be published.
5. It is troubling that all molecular identity in the study, discussed and represented in the figures, all derives from mass spectroscopy. Even stereochemistry is depicted in 4-AmcB and 5-AmcB, which cannot possibly be determined by mass. Without proper molecular analysis,

by NMR or comparable methodology, this reviewer is not convinced of these structures.

Reviewer #2 (Remarks to the Author):

In the review written by Kuzuyama and coworkers the authors report on “Unprecedented type II polyketide synthases-like enzymes that expand natural product diversity”

The manuscript is written well and I cannot see any formal reasons why this manuscript should not be acceptable for publication. However, my enthusiasm was fading away when I was evaluating the manuscript and I recommend a few corrections.

1. The abstract is very dry and somewhat undersells the findings of the study. After reading it, I don't really know why I should care about this study and what's the knowledge gap / biological question you are addressing?
2. The introduction part is somehow misleading. It is a bit unclear where the introductory part ends and where the results summary starts. Please re-write.
3. Figure 4: I can follow the logic for conversion of compound 1 to compound 2 because I am now, after several rounds of reading, familiar with this pathway. However, the logic might not be entirely clear to the general reader. The entire figure needs to be improved.
4. Figure 5: What is the main message of this figure?

Minor points:

Please show the structure of BAY 10-8888.

Reviewer #3 (Remarks to the Author):

The authors describe the synthesis of Cispentacin, a non-proteinogenic, anti-fungal amino acid by using seven recombinant proteins either in a heterologous host or, more importantly, in-vitro. The manuscript is well written with many details on the molecular interplay between the seven proteins needed for this biosynthetic pathway. This detailed analysis allowed the identification of a new subfamily of type II Polyketide synthases (PKS).

Due to my background, I will focus on the computational aspect of this work.

While being already close to self-explanatory, I think some readers might benefit from adding Table legends. For example, in Table S1, it would be helpful to understand how exactly you

derived E-values (tool, version, parameters, database you searched against etc.). This is crucial for reproducing your results. Alternatively, you could also think about adding those details in a dedicated section in Methods and re-direct readers there.

Similarly, for reproducing the ColabFold/AlphaFold2-analysis (Fig. S15-S17), it would be very helpful to have some more details on the in-silico experiments, i.e., which version of ColabFold was used, which database was searched for generating MSAs, did you use amber-relaxation and/or templates or more generally, which parameters were used for prediction (if all default, please, make this point explicit to avoid ambiguity), which of the five models output by ColabFold/AlphaFold2 did you use, and most importantly, what was the quality-/reliability-score output for the predictions, e.g., which pLDDT/pTM was output? See recent work of Arne Elofsson as a guideline for interaction reliability thresholds: 10.1038/s41594-022-00910-8. - Similar to the comment above, consider adding a dedicated section for these details in Methods.

You mention that the interaction was “similar to those of the typical KS-CLF complexes” (p. 6, l. 11). Consider adding a PDB-ID as reference to give readers some guidance on what exactly you compare (you already do this in the Fig. caption of Fig S15 but adding the ID to main text might help some readers). You could even quantify structural similarity between an experimentally measured interface/active centre and the predicted one (see comment below on structural similarity).

Expanding your bioinformatics analysis from sequence-search to structure-search might reveal interesting novelties as you can detect similarity at much lower levels. Recent advances (foldseek - 10.1101/2022.02.07.479398 - available via <https://search.foldseek.com>) allow structure search even at the scale of UniProt/AlphaFold-DB. I think such an analysis might complement the existing sequence search and potentially detect novel, very remote homologs to expand the analysis shown in Fig. 5A. This might also help to unravel the function of AmcG. Similarly, even the current analysis could benefit strongly from quantifying structural similarity (e.g. adding TM-score/LDDT in Fig. 1B). While I would understand if the authors considered adding an analysis on detecting new remote homologs via foldseek to be beyond this work, I think quantifying structural similarity between the proteins they already found would strengthen the message of the manuscript.

Protein language models were shown to be able to detect functional and structural similarity beyond sequence similarity (goPredSim - 10.1038/s41598-020-80786-0 - available via <https://embed.predictprotein.org/>). It might be worth checking function prediction in the form of Gene Ontology terms for e.g. AmcG. Again, this comment is rather meant as a suggestion to further improve the manuscript but I would understand if the authors would consider this analysis to be beyond the scope of their current work.

Fig. S14: please, give some more details on how you derived the secondary structure, e.g., by providing PDB- or AlphaFoldDB-IDs. Consider quantifying conservation by cross-checking related PDB-IDs in consurf-DB (<https://consurfdb.tau.ac.il/>) or predicted conservation scores (<https://embed.predictprotein.org/>).

Fig. S15: consider using a different background colour than black (e.g. white).

Fig. S15: add length of the AmcF and AmcG to PAE in panel A.

Fig. S16B: always indicate whether you computed the structure of a complex from which you extracted one protein for further analysis or whether you predicted the protein(s) separately, i.e., in isolation. Comparing the predicted structures of the individual proteins in isolation and when bound in complexes might also reveal interesting structural shifts upon binding (again: rather curiosity, and ok if considered to be beyond this work).

Fig. S16: explain why you use two different reference PDB entries for the KS-CLF heterodimer complex (6qsp vs 6kxd). Probably there is a good reason but I did not immediately see the motivation behind this choice. This comparison could benefit from structural similarity quantification via TM-score, LDDT, RMSD etc.

Fig. S16 A-C (and Fig. 15B): there are grey boxes around some figures. Maybe this is an artefact of my PDF rendering but double check to avoid this in the final version.

Fig. S16: add small arrows indicating rotation direction and angle of rotation.

Fig. S17C: similar to comments above, the manuscript could benefit from quantifying structural similarity and giving details on how ColabFold was run.

Reviewer #4 (Remarks to the Author):

This review covers the manuscript entitled “Unprecedented Type II Polyketide Synthase-like Enzymes that Expand Natural Product Diversity” by Hibi et al. submitted to the Nature Communications. The manuscript reports the production of cispentacin in *S. albus*, and identification of biosynthetic pathway for the five-membered nonaromatic skeleton of cispentacin. Reported experimental data are appropriate and appear well done. However, following few points need to be elaborated to support the valuable finding:

1. Authors reported the production of cispentacin in *S. albus* G153 (cispentacin nonproducing bacteria). Did authors try cispentacin production in any other bacteria (cispentacin nonproducing bacteria) by the introduction of *amcB-amcH* genes? I believe that this type of verification experiments will further support these findings.

2. Authors stated that “introduction of the *amcB-amcH* genes into a cispentacin nonproducing

S. albus G153 leads to the production of cispentacin or its biosynthetic intermediate(s). The resulting *S. albus* G153/pSEamcB-H transformant harboring amcB-H produced an average of 200 mg/L cispentacin, demonstrating that the seven gene regions spanning amcB to amcH are responsible for cispentacin biosynthesis". Did the authors try to change the sequence of seven gene region (i.e. amcB-amcH) and expressed in in *S. albus* G153 or miss any gene and express the altered region in *S. albus* G153 and observe reduced or no production of cispentacin?

3. Information on the statistical analyses used are missing, especially related to the quantification of cispentacin.

4. Figure S6. SDS-PAGE analysis of purified recombinant proteins: there are many bands of proteins that need to be explained. Such as in Lane B, there are two additional bands other than AmcB, calculated (~ MW 12 kDa). Similarly, in Lanes D, E, FG.

5. References need to be presented in same format.

Reviewer #1 (Remarks to the Author):

The manuscript “Unprecedented Type II Polyketide Synthase-like Enzymes that Expand Natural Product Diversity” by Hibi, et al., describes studies into the biosynthesis of cispentacin from amipurimycin producing streptomycetes. Fascinatingly, it appears that this 5-membered cyclic β -amino acid is prepared from α -ketoglutarate from the action of a collection of type II PKS enzymes, including ACP, AT, KS, and CLF-like stand-alone domains. This work has the potential to dramatically revise our definitions of type II PKS enzymes, which had already been recently revised through the elucidation of polyene PKS pathways. Despite the uniqueness of these studies, the manuscript contains significant conceptual and experimental deficiencies that must be corrected and/or completed before this study can be sufficient for publication. The following major items should be addressed before further determination:

Thank you for appreciating the uniqueness of our research.

We also thank the reviewer for their constructive comments.

1. Page 4. Line 12. “It was also confirmed that the resulting holo-AmcB was indeed converted to malonyl-AmcB (11767 Da) in the presence of malonyl-CoA in vitro (Figure 14 S8).” This claimed result is an artifact of acyl transfer that occurs in in vitro experimentation that has now been very well documented. This spontaneous acylation cannot serve as the physiological malonyltransfer mechanism, as it is likely that other CoA species (such as acetyl-CoA) will predominate in vivo.

Thank you for pointing this out.

As this reviewer mentions, this spontaneous malonyl transfer could be an artifact; therefore, we have revised the manuscript to emphasize this point. Cfa1, the AmcB homolog, has been examined in previous studies, which demonstrated that FabD is the proper enzyme for spontaneous malonyl transfer. Therefore, we have added the following sentences (Page 4, Line 13) and cited the corresponding article [12]. “The malonyl transfer to ACP has been validated with the AmcB homolog Cfa1, and it was documented that FAS malonyl CoA:ACP transacylase (FabD), which is recruited to Cfa1, catalyzes the reaction [12]. Thus, with respect to AmcB, the malonyl transfer to AmcB may also be catalyzed by FabD, which is presumably recruited to AmcB.”

2. Page 4, line 23. “AmcH is the first adenylate-forming acyltransferase (AT) that forms 2-

oxoglutaryl-ACP." The use of acyltransferase (AT) nomenclature is incorrect. ATs are CoA dependent enzymes, whereas were the enzyme is shown to be ATP dependent. This classifies it as an adenylating enzyme (acyl-AMP forming), abbreviated as A.

Thank you. We have revised the manuscript as suggested (Page 4, Line 25).

3. Creating the new nomenclature "cyclization factor (CYF)" is a new concept and should be evaluated by all reviewers. If the proposed mechanism is fully demonstrated, I approve of the new name.

Thank you for commenting on the nomenclature "cyclization factor (CYF)". We believe that all reviewers accept the nomenclature because no other reviewers commented on the subject, and we confirmed the function of AmcF-AmcG in the revised manuscript by demonstrating the structure of the reaction product (see also 5).

4. Page 7 Line 19-34. Again, this study is dependent upon a spontaneous reaction, here reduction of 3-AmcB to 4-AmcB by NADH. While such an outcome may be visualized for in vitro experimentation, it is inconceivable that this could be the physiological mechanism. The proper enzyme that catalyzes this reaction must be identified. Could it be FabI, or perhaps there is an adjacent reductase in the operon? The authors must conduct this due diligence before this study can be published.

In accordance with the reviewer's comments, we attempted to identify the enzyme that catalyzes the reduction of 3-AmcB to 4-AmcB in the presence of NADH. Eventually, we demonstrated that both eFabI derived from *E. coli* and sFabI derived from *Streptomyces albus* G153 catalyze the reduction of 3-AmcB to 4-AmcB with NADH (Page 9, Line 21). As fatty acid synthase (FAS) forms a multienzyme system with KS, AT, KR, DH, ER and ACP in prokaryotes, we suspect that AmcB ACP recruits FabI reductase. We have added Figure S22 to illustrate a series of experimental results and the experimental method (Page 18, Line 46). We greatly appreciate the reviewer's comments, which have improved the value of this study.

5. It is troubling that all molecular identity in the study, discussed and represented in the figures, all derives from mass spectroscopy. Even stereochemistry is depicted in 4-AmcB and 5-AmcB, which cannot possibly be determined by mass. Without proper molecular analysis, by NMR or comparable methodology, this reviewer is not convinced of these structures.

To provide convincing evidence for the readers and reviewers, we chemically synthesized 3-oxocyclopent-1-ene-1,2-dicarboxylic acid (Compound **2**) and verified its identity with the enzyme product intermediate 2 (Page 5, Line 4, Fig. 2C-D). The structures of Compounds 3, 4, and 5 can be determined based on the functions of the enzymes that catalyzes each reaction and the HR mass spectrometry results. Therefore, we believe this reviewer will be convinced of these structures.

Regarding stereochemistry, first, the stereochemistry of cispentacin biosynthesized by the amipurimycin biosynthesis (amc) genes is identical to that of the cispentacin moiety of amipurimycin. In addition, through comparison with the cispentacin standard, it was determined that cispentacin biosynthesized by the amc genes has structural support.

In the final step of cispentacin biosynthesis, cispentacin is simply released from cispentacin-AmcB by the thioesterase enzyme AmcD. Therefore, the stereochemistry of the cispentacin moiety of cispentacin-AmcB should be identical to that of cispentacin. Since AmcC is an aminotransferase and only stereoselectively introduces an amino group to the keto group of Intermediate 4, the stereochemistry of Intermediate 4 is also identical to that of cispentacin. Thus, we proposed a biosynthetic pathway for cispentacin, including the stereochemistry of all the intermediates (Figure 4). We have added the above sentences to describe the stereochemistry of the intermediates in cispentacin biosynthesis (Page 14, Line 12).

Since the stereochemistry of Intermediate 4 was determined, we can discuss the stereoselectivity of the NADH-dependent enoyl reduction of Intermediate 3 by FabI. We have added some sentences to describe the stereoselectivity of FabI in the Discussion section (Page 15, Line 16).

Reviewer #2 (Remarks to the Author):

In the review written by Kuzuyama and coworkers the authors report on “Unprecedented typeII polyketide synthases-like enzymes that expand natural product diversity”

The manuscript is written well and I cannot see any formal reasons why this manuscript should not be acceptable for publication. However, my enthusiasm was fading away when I was evaluating the manuscript and I recommend a few corrections.

Thank you for your evaluation of the paper.

We also thank the reviewer for their constructive comments.

1. The abstract is very dry and somewhat undersells the findings of the study. After reading it, I don't really know why I should care about this study and what's the knowledge gap / biological question you are addressing?

We have completely revised the abstract. We believe that the revised abstract is much better than the previous abstract.

2. The introduction part is somehow misleading. It is a bit unclear where the introductory part ends and where the results summary starts. Please re-write.

As suggested, we have removed the confusing sentence from the introduction and have added a new sentence on the potential of Type II PKS (Page 1, L30-32). We believe that this revision eliminates the misleading results.

3. Figure 4: I can follow the logic for conversion of compound 1 to compound 2 because I am now, after several rounds of reading, familiar with this pathway. However, the logic might not be entirely clear to the general reader. The entire figure needs to be improved.

We have revised the figure and improved the figure legend so that general readers can understand the logic for the conversion of Compound 1 to Compound 2.

4. Figure 5: What is the main message of this figure?

The message is that many unidentified natural products are presumably related to the biosynthetic mechanisms elucidated in this work and that the analysis of these BGCs could lead to the identification of novel natural products. This message is described in the section "*amcG* homolog-containing BGCs are widespread across several bacterial phyla". We believe that the readers will understand these statements.

Minor points:

Please show the structure of BAY 10-8888.

We have added the structure of BAY 10-8888 (Fig. 1).

Reviewer #3 (Remarks to the Author):

The authors describe the synthesis of Cispentacin, a non-proteinogenic, anti-fungal amino acid by using seven recombinant proteins either in a heterologous host or, more importantly, in-vitro. The manuscript is well written with many details on the molecular interplay between the seven proteins needed for this biosynthetic pathway. This detailed analysis allowed the identification of a new subfamily of type II Polyketide synthases (PKS).

Due to my background, I will focus on the computational aspect of this work.

We thank the reviewer for their constructive comments.

Your points regarding the computational aspects are very informative and meaningful for us.

While being already close to self-explanatory, I think some readers might benefit from adding Table legends. For example, in Table S1, it would be helpful to understand how exactly you derived E-values (tool, version, parameters, database you searched against etc.). This is crucial for reproducing your results. Alternatively, you could also think about adding those details in a dedicated section in Methods and re-direct readers there.

Thank you for your suggestions.

We have added the E-values and the tool, version, parameters, and database in Table S1 as you suggested.

Similarly, for reproducing the ColabFold/AlphaFold2-analysis (Fig. S15-S17), it would be very helpful to have some more details on the in-silico experiments, i.e., which version of ColabFold was used, which database was searched for generating MSAs, did you use amber-relaxation and/or templates or more generally, which parameters were used for prediction (if all default, please, make this point explicit to avoid ambiguity), which of the five models output by ColabFold/AlphaFold2 did you use, and most importantly, what was the quality-/reliability-score output for the predictions, e.g., which pLDDT/pTM was output? See recent work of Arne Elofsson as a guideline for interaction reliability thresholds: 10.1038/s41594-022-00910-8. - Similar to the comment above, consider adding a dedicated section for these details in Methods.

We have added an explanation of the ColabFold method in the Methods section (Page 20, Line 38), as you suggested.

Thank you for referring us to Arne Elofsson's study.

We have validated the reliability of the AmcF–G interaction based on that study, which shows

that the complex structure is reliable. (Page 7, Line 14)

You mention that the interaction was “similar to those of the typical KS-CLF complexes” (p. 6, l. 11). Consider adding a PDB-ID as reference to give readers some guidance on what exactly you compare (you already do this in the Fig. caption of Fig S15 but adding the ID to main text might help some readers). You could even quantify structural similarity between an experimentally measured interface/active centre and the predicted one (see comment below on structural similarity).

Yes, we have revised the manuscript as suggested (Page 7, Line 13).

To quantify the structural similarity between an experimentally measured interface/active center and the predicted interface/active center, we provided the RMSD values for the superposition of the AmcF–AmcG structure with the experimentally measured KS–CLF structure in PyMOL in Figures S15E and S16A. Could you see them?

Expanding your bioinformatics analysis from sequence-search to structure-search might reveal interesting novelties as you can detect similarity at much lower levels. Recent advances (foldseek - 10.1101/2022.02.07.479398 - available via <https://search.foldseek.com>) allow structure search even at the scale of UniProt/AlphaFold-DB. I think such an analysis might complement the existing sequence search and potentially detect novel, very remote homologs to expand the analysis shown in Fig. 5A. This might also help to unravel the function of AmcG. Similarly, even the current analysis could benefit strongly from quantifying structural similarity (e.g. adding TM-score/LDDT in Fig. 1B). While I would understand if the authors considered adding an analysis on detecting new remote homologs via foldseek to be beyond this work, I think quantifying structural similarity between the proteins they already found would strengthen the message of the manuscript.

Thank you for your very useful suggestions.

We used foldseek to search for new remote homologs and found several putative AmcG homologs in our search against AlphafoldDB. However, we did not describe them in the text because we did not find results with greater advantages over the sequence search results.

Protein language models were shown to be able to detect functional and structural similarity beyond sequence similarity (goPredSim - 10.1038/s41598-020-80786-0 - available via <https://embed.predictprotein.org/>). It might be worth checking function prediction in the

form of Gene Ontology terms for e.g. AmcG. Again, this comment is rather meant as a suggestion to further improve the manuscript but I would understand if the authors would consider this analysis to be beyond the scope of their current work.

Thank you for pointing out the new tool.

Although we attempted to predict the function of AmcG using the tool, unfortunately, we were unable to obtain function prediction that should be added, so we did not describe it in the text.

Fig. S14: please, give some more details on how you derived the secondary structure, e.g., by providing PDB- or AlphaFoldDB-IDs. Consider quantifying conservation by cross-checking related PDB-IDs in consurf-DB (<https://consurfdb.tau.ac.il/>) or predicted conservation scores (<https://embed.predictprotein.org/>).

Yes, we have revised the caption as suggested (in the Fig. caption of Fig. S14).

Fig. S15: consider using a different background colour than black (e.g., white).

Yes, we have revised the figure as suggested (Fig. S15).

Fig. S15: add length of the AmcF and AmcG to PAE in panel A.

Yes, we have revised the figure as suggested (Fig. S15).

Fig. S16B: always indicate whether you computed the structure of a complex from which you extracted one protein for further analysis or whether you predicted the protein(s) separately, i.e., in isolation. Comparing the predicted structures of the individual proteins in isolation and when bound in complexes might also reveal interesting structural shifts upon binding (again: rather curiosity, and ok if considered to be beyond this work).

Yes, we have revised the caption as suggested (in the Fig. caption of Fig. S16).

Thank you for the structure proposal. However, our analysis to date has not revealed any noticeable structural shift that should be added between the predicted structures of the individual proteins in isolation and when bound in complexes.

Fig. S16: explain why you use two different reference PDB entries for the KS-CLF heterodimer complex (6qsp vs 6kxd). Probably there is a good reason but I did not

immediately see the motivation behind this choice. This comparison could benefit from structural similarity quantification via TM-score, LDDT, RMSD etc.

6kxd was used because there is a triplet structure (6kxf) including ACP, so information on the substrate binding site can be obtained through comparing these structures. On the other hand, 6qsp was used because it is the smallest structure in type II CLFs for which a crystal structure occurs.

Since CLFs exhibit a variety of sizes, we thought that by comparing the smallest structure of CLFs with CYFs, we could highlight the structural homology and differences with CLFs.

Fig. S16 A-C (and Fig. 15B): there are grey boxes around some figures. Maybe this is an artefact of my PDF rendering but double check to avoid this in the final version.

To remove the unnecessary gray boxes, we have revised the figures as suggested (Fig. S15 and Fig. S16).

Fig. S16: add small arrows indicating rotation direction and angle of rotation.

Yes, we have revised the figure as suggested (Fig. S16).

Fig. S17C: similar to comments above, the manuscript could benefit from quantifying structural similarity and giving details on how ColabFold was run.

As mentioned above, a description of the ColabFold method has been added to the Methods section (Page 20, Line 38).

Reviewer #4 (Remarks to the Author):

This review covers the manuscript entitled “Unprecedented Type II Polyketide Synthase-like Enzymes that Expand Natural Product Diversity” by Hibi et al. submitted to the Nature Communications. The manuscript reports the production of cispentacin in *S. albus*, and identification of biosynthetic pathway for the five-membered nonaromatic skeleton of cispentacin. Reported experimental data are appropriate and appear well done. However, following few points need to be elaborated to support the valuable finding:

1. Authors reported the production of cispentacin in *S. albus* G153 (cispentacin nonproducing bacteria). Did authors try cispentacin production in any other bacteria (cispentacin nonproducing bacteria) by the introduction of *amcB-amcH* genes? I believe that this type of verification experiments will further support these findings.

Streptomyces genes are generally difficult to express in bacteria other than Streptomyces because of their high GC content and specialized promoter and SD sequences. Therefore, we believe that introducing the *amcB-amcH* genes into bacteria other than Streptomyces would not yield any new findings. We introduced the *amcB-amcH* genes into *E. coli*, but it did not produce cispentacin.

In addition, because we demonstrated the reconstitution of cispentacin production in vitro as well, we do not think that the experiments you suggested would provide new insights.

2. Authors stated that “introduction of the *amcB-amcH* genes into a cispentacin nonproducing *S. albus* G153 leads to the production of cispentacin or its biosynthetic intermediate(s). The resulting *S. albus* G153/pSE*amcB-H* transformant harboring *amcB-H* produced an average of 200 mg/L cispentacin, demonstrating that the seven gene regions spanning *amcB* to *amcH* are responsible for cispentacin biosynthesis”. Did the authors try to change the sequence of seven gene region (i.e. *amcB-amcH*) and expressed in *S. albus* G153 or miss any gene and express the altered region in *S. albus* G153 and observe reduced or no production of cispentacin?

No, we did not try to change the sequence of the seven gene regions and expressed in *S. albus* G153 or delete any gene and express the altered region in *S. albus* G153. Instead, we demonstrated that seven enzymes are essential for cispentacin production in vitro. Therefore, if any of the seven genes is missing, cispentacin production is impossible.

3. Information on the statistical analyses used are missing, especially related to the quantification of cispentacin.

We originally described our quantification methods in the Methods section and SI. This reviewer may have missed the section. We have added a standard deviation for the values of cispentacin production to Figure S5.

4. Figure S6. SDS-PAGE analysis of purified recombinant proteins: there are many bands of proteins that need to be explained. Such as in Lane B, there are two additional bands other

than AmcB, calculated (~ MW 12 kDa). Similarly, in Lanes D, E, FG.

We have added the following sentences to the figure legend of Figure S6.

“In Lane B, a small amount of phosphopantetheinyl transferase (25 kDa), which is coexpressed in *E. coli* cells for phosphopantetheinyl transfer of AmcB, was also detected. In Lanes C to H, each target protein marked with an asterisk is major, although unknown trace proteins are detected.”

5. References need to be presented in same format.

We have revised the references and presented them in the same format in the revised manuscript.

Reviewer #1 (Remarks to the Author):

Most of the comments have been responded to properly, however, one item remains problematic. The response for R1, Item 5, "Since AmcC is an aminotransferase and only stereoselectively introduces an amino group to the keto group of Intermediate 4, the stereochemistry of Intermediate 4 is also identical to that of cispentacin" is not accurate. While reaction 4 appears to be an aminotransferase reaction, you cannot assume that the stereochemistry is that of cispentacin without experimental evidence. This is not an arbitrary or difficult request -- it can be easily verified by chiral derivatizing agents (DOI: 10.1007/978-1-61779-445-2_7) and should be validated.

Reviewer #3 (Remarks to the Author):

Thank you for your thorough revision. All my previous points were addressed adequately

Reviewer #4 (Remarks to the Author):

This re-review covers the manuscript entitled "Unprecedented Type II Polyketide Synthase-like Enzymes that Expand Natural Product Diversity" by Hibi et al. submitted to Nature Communications after covering some logical points. The manuscript reports the production of cispentacin in *S. albus*, and identification of biosynthetic pathway for the five-membered nonaromatic skeleton of cispentacin. The authors now answered and added justification to all objections/ confusions. I would like to say that the reported data are appropriate and appear well, hence fine for publication.

Reviewer #1 (Remarks to the Author):

Most of the comments have been responded to properly, however, one item remains problematic. The response for R1, Item 5, "Since AmcC is an aminotransferase and only stereoselectively introduces an amino group to the keto group of Intermediate 4, the stereochemistry of Intermediate 4 is also identical to that of cispentacin" is not accurate. While reaction 4 appears to be an aminotransferase reaction, you cannot assume that the stereochemistry is that of cispentacin without experimental evidence. This is not an arbitrary or difficult request -- it can be easily verified by chiral derivatizing agents (DOI: 10.1007/978-1-61779-445-2_7) and should be validated.

Thank you for pointing this out and suggesting a useful method.

According to the comments, we performed additional experiments with the chiral derivatizing agents L-FDLA and D-FDLA. The method was added to the Methods section (Page 21, Line 50–52 & Page 22, Line 1–25). Relevant references have also been added (Page 25–26, Page 69).

The experimental results showed that the stereochemistry of intermediate **5** is identical to that of the final product cispentacin, demonstrating that AmcC catalyzes stereoselective transamination to the keto group at C2 of **4**, yielding (2*S*)-stereochemistry of **5** (Page 9, Line 27–31; Page 14, Line 14–17 & Page 14, Line 1 ; Fig. S24).

We are pleased that the reviewer's precise comments have removed a concern about stereochemistry and increased the value of this study. We would like to thank the reviewer again.

Other corrections

We have added an important description regarding AmcE (Page 8, Line 6–8).

Reviewer #1 (Remarks to the Author):

All items have been addressed.

I would like to congratulate the authors on a very nice manuscript!